# Tuning the Cytotoxicity of Bis-Phosphino-Amines Ruthenium(II) Para-Cymene Complexes for Clinical Development in Breast Cancer

**DOI:** 10.3390/pharmaceutics13101559

**Published:** 2021-09-26

**Authors:** Elena Domínguez-Jurado, Francisco J. Cimas, José Antonio Castro-Osma, Alberto Juan, Agustín Lara-Sánchez, Antonio Rodríguez-Diéguez, Alexandr Shafir, Alberto Ocaña, Carlos Alonso-Moreno

**Affiliations:** 1Centro Regional de Investigaciones Biomédicas, Unidad NanoCRIB, 02008 Albacete, Spain; elena.dominguez@uclm.es (E.D.-J.); joseantonio.castro@uclm.es (J.A.C.-O.); alberto.juan@uclm.es (A.J.); 2Facultad de Farmacia de Albacete, Universidad de Castilla-La Mancha, 02008 Albacete, Spain; 3Oncología Traslacional, Unidad de Investigación del Complejo Hospitalario Universitario de Albacete, 02008 Albacete, Spain; franciscojose.cimas@uclm.es; 4Facultad de Ciencias y Tecnologías Químicas, Universidad de Castilla-La Mancha, 13005 Ciudad Real, Spain; Agustin.Lara@uclm.es; 5Departamento de Química Inorgánica, Facultad de Ciencias, Universidad de Granada, Avda de Fuentenueva s/n, 18071 Granada, Spain; antonio5@ugr.es; 6Department of Biological Chemistry, Institute of Advanced Chemistry of Catalonia, IQAC-CSIC, c/Jordi Girona 18-26, 08034 Barcelona, Spain; ashafir@iciq.es; 7Experimental Therapeutics Unit, Hospital Clínico San Carlos, IdISSC and CIBERONC, 28040 Madrid, Spain

**Keywords:** breast cancer, metallodrugs, RAPTA derivatives, phosphino-amine ligands

## Abstract

Despite some limitations such as long-term side effects or the potential presence of intrinsic or acquired resistance, platinum compounds are key therapeutic components for the treatment of several solid tumors. To overcome these limitations, maintaining the same efficacy, organometallic ruthenium(II) compounds have been proposed as a viable alternative to platinum agents as they have a more favorable toxicity profile and represent an ideal template for both, high-throughput and rational drug design. To support the preclinical development of bis-phoshino-amine ruthenium compounds in the treatment of breast cancer, we carried out chemical modifications in the structure of these derivatives with the aim of designing less toxic and more efficient therapeutic agents. We report new bis-phoshino-amine ligands and the synthesis of their ruthenium counterparts. The novel ligands and compounds were fully characterized, water stability analyzed, and their in vitro cytotoxicity against a panel of tumor cell lines representative of different breast cancer subtypes was evaluated. The mechanism of action of the lead compound of the series was explored. In vivo toxicity was also assessed. The results obtained in this article might pave the way for the clinical development of these compounds in breast cancer therapy.

## 1. Introduction

Cancer is one of the most devastating diseases in the world and particularly breast cancer is a leading cause of death in women [1]. Classical treatments for this disease include chemotherapy, hormonotherapy, and targeted agents such as anti-HER2 treatments [1]. However, it is a global concern that many patients harbor an inherent resistance to these drugs, and even for those that respond, progression after a specific period of time becomes frequent [2,3].

Breast cancer can be divided into three different subtypes based on their genomic/transcriptomic profile, the HER2 enriched, the basal-like, and the luminal breast [4,5]. The triple-negative breast cancer subtype mimics the basal-like subgroup and is termed in that way due to the lack of HER2 and estrogen receptor expression, representing 15% of breast tumors. It also harbors a more aggressive behavior along with a limited number of efficient therapeutic options [5]. In this regard, the identification of novel targets with potential for druggability, the optimization of those that are currently in clinical use, or the development of new compounds with higher efficacy and a safer toxicity profile, is the main objective.

Platinum-based therapies including carboplatin and cisplatin have been the standard treatment of care as a chemotherapy backbone, in many solid tumors, such as in triple-negative and HER2 positive breast cancer [6,7,8]. Platinum agents are highly cytotoxic and unspecific, therefore, they can produce severe undesired side effects [9]. In addition, some tumors express an inherent resistance to these agents and others acquired secondary resistance after being exposed to these agents for a period of time [2]. In this context, other metallic compounds with better biological and pharmacological properties are needed. Rutheniums have emerged as a novel and promising anticancer family of agents [10,11]. Ruthenium-based therapy might overcome platinum resistance, and improve the narrow therapeutic window of ‘the platinum family of agents’ [12,13]. In addition, ruthenium compounds have been shown to mediate their biological effect through a combination of anti-proliferative activity, anti-angiogenic action, and an anti-metastatic property which can clearly improve the action of current platinum derivatives [14,15,16].

To date, there have been several ruthenium compounds in clinical trials [17]. NAMI-A was the first ruthenium agent that eventually reached the clinical stage giving rise to high expectations for this new class of metal-based anticancer drugs [15,18]. KP1019 and its sodium salt KP1339 were the following compounds to be evaluated [19]. NAMI-A and KP compounds were tested in animal models demonstrating activity. Phase II combination studies of NAMI-A along with gemcitabine against lung cancer reported moderate clinical activity limiting their further evaluation [20]. Phase I clinical investigation of KP1019 for patients with advanced solid tumors showed a moderate antitumor activity [21]. The ruthenium compound TLD1433 completed phase I clinical trials for intravesical photodynamic therapy in patients with a non-muscle invasive bladder tumor and is currently recruiting patients in a phase II study [22,23].

Organometallic ruthenium compounds such as RAPTA-C [24] and RM175 [25,26] are in advanced preclinical studies. In this regard, a very high number of new organometallic ruthenium agents have been reported as potential candidates for clinical translation. Many examples of the screening of mono-, di-, and poly-nuclear ruthenium compounds with biologically inactive and active auxiliar ligands to give chiral and achiral complexes have been reported by advanced biological methods to understand their mechanism of action [27,28,29,30,31,32,33,34]. In this context, successful ruthenium complexes have been reported for the treatment of breast cancer [35]. In this regard, we proposed bis-phoshino-amines as versatile auxiliar ligands for a direct and simple generation of RAPTA-C derivatives [36]. The cytotoxicity of the novel ruthenium compounds was studied in a panel of tumoral cells to exploit them as a promising family of therapeutic agents. In addition, the lead compound of the series was selected for further evaluation (Figure 1). In a second study, we focused our efforts on understanding its mode of action, pharmacokinetic and biodistribution parameters and proposed plausible targets to improve subsequent designs [37].

Herein, a new family of ligands was synthetized and characterized to generate their ruthenium counterparts. The novel ruthenium agents were evaluated against a panel of breast tumoral cells representative of the three described breast cancer subtypes and were also compared to platinum agents. From this work, a new lead complex was obtained, and its mechanism of action was studied in depth. Finally, in vivo toxicity of the lead compound and its versatility against other tumor cells overexpressing HER2+ was performed.

## 2. Materials and Methods

### 2.1. General Procedure

Synthesis reactions were performed using standard Schlenk and glove-box techniques under an atmosphere of dry nitrogen. CH_2_Cl_2_ and hexane were pre-dried over CaCl_2_ and sodium wire, respectively, and distilled under nitrogen from sodium-potassium alloy (hexane), and CaCl_2_ (CH_2_Cl_2_). CDCl_3_, DMSO-d^6^, and D_2_O were stored over activated 4 Å molecular sieves and degassed by several freeze-thaw cycles. All NMR experiments were conducted in deuterated solvents at 297 K in a Varian FT-400 spectrometer (VARIAN Inc., Palo Alto, California, USA) equipped with a 4 nucleus ASW PFG ^1^H/^19^F/^13^C/{^15^N-^31^P}. The ^1^H π/2 pulse length was adjusted for each sample. ^1^H- and ^13^C{^1^H}-NMR chemical shifts (δ) are given in ppm relative to TMS. ^31^P{^1^H}-NMR chemical shifts are given in ppm relative to H_3_PO_4_ (85%). Coupling constants (*J*) are documented in Hz. The solvent signals were used as references and chemical shifts converted to the TMS scale. IR experiments were conducted on an FT/IR-4000 Series Jasco Instruments (Jasco Analytics, Madrid, Spain). The UV-Vis absorption spectra were recorded at room temperature using a Cary 100 spectrophotometer (Agilent, Madrid, Spain) using a slit width of 0.4 nm and a scan rate of 600 nm/min. Elemental Analysis was performed at the Unidad de Análisis Químico Elemental, University complutense, Madrid (Spain) using an Elementary Chemical Analyzer LECO CHNS-932. Mass spectroscopy was performed at the Unidad de Espectrometría de Masas, University complutense, Madrid (Spain) using a MAXXIS II Bruker spectrometer.

### 2.2. Synthesis and Characterization of Ligands L1–L10

To a solution of chlorodiphenylphosphine (0.54 mL, 2.92 mmol) in toluene (15 mL) at 0 °C were added dropwise triethylamine (0.45 mL, 3.22 mmol) and the corresponding amine (1.45 mmol). After a 30 min stirring time, the ice bath was removed, and the stirring continued. Monitoring by NMR showed all starting materials to have been consumed within 5 h; at this point, the solution was filtered to remove the Et_3_NHCl formed. On removing all volatiles under reduced pressure, the resultant white solid was repeatedly washed with degassed MeCN until no Ph_2_PP(O)Ph_2_ impurity could be detected by ^31^P{^1^H}-NMR.

*N-tert-butyl-N-(diphenylphosphanyl)-1,1-diphenylphosphanamine (L1)*: After extracting in toluene, *L1* was recovered as a white solid with a yield of 86%. ^1^H NMR (400 MHz, CDCl_3_) δ 7.88–7.17 (m, 20 H, Ar), 1.43 (s, J = 7.9 Hz, 9 H, CH_3_). ^13^C{^1^H}-NMR (101 MHz, CDCl_3_) *δ* 140.74 (4 C, Ar quaternary (PPh_2_)), 132.81 (8 C, o-Ar CH (PPh_2_)), 128.23 (8 C, m-Ar CH(PPh_2_)), 127.91 (4 C, p-Ar CH(PPh_2_)), 63.8 (1 C, C–(CH_3_)_3_), 32.75 (3 C, CH_3_). ^31^P NMR (162 MHz, CDCl_3_) *δ* 21.39. UV-vis: maximum absorbance at 288 nm. IR: 2961 cm^−1^ (C–H sp^3^ stretching), 1476–1431 cm^−1^ (two bands C=C aromatic stretching), 1174 cm^−1^ (C–N stretching), 1087 cm^−1^ (C–P stretching). Elemental analysis calcd (%) for C_28_H_29_NP_2_: C, 76.17; H, 6.62; N, 3.17; P, 14.03; found: C, 76.12; H, 6.52; N, 3.31.

*N-(diphenylphosphanyl)-1,1-diphenyl-N-propylphosphanamine (L2):* After extracting in toluene, L3 was recovered as a white solid with a yield of 84%. ^1^H NMR (400 MHz, CDCl_3_) δ 7.40–7.33 (m, 8H, *o*-Ar), 7.30–7.24 (m, 12H, *m*- and *p*-Ar), 3.20–3.09 (m, 2H, *-CH_2_-*CH_2_-CH_3_), 1.11–0.95 (m, 2H, –CH_2_–*CH_2_*–CH_3_), 0.45 (t, J = 7.3 Hz, 3H, –CH_2_–CH_2_–*CH_3_*). ^13^C{^1^H}-NMR (101 MHz, CDCl_3_) δ 139.66 (4 C, Ar quaternary (PPh_2_)), 132.65 (8 C, *o*-Ar CH (PPh_2_)), 128.65 (6 C, *m-* and *p-*Ar CH (PPh_2_)), 127.95 (6 C, *m-* and *p-*Ar CH (PPh_2_)), 54.65 (2 C, –*CH_2_*–CH_2_–CH_3_), 24.60 (2 C, –CH_2_–*CH_2_*–CH_3_), 10.94 (3 C, –CH_2_–CH_2_–*CH_3_*). UV-vis: maximum absorbance at 284 nm. IR: 2957 cm^−1^ (C–H sp^3^ stretching), 1478–1433 cm^−1^ (two bands C=C aromatic stretching), 1090 cm^−1^ (C–P stretching), 1057 cm^−1^ (C–N stretching). Elemental analysis calcd (%) for C_27_H_27_NP_2_: C, 75.86; H, 6.37; N, 3.28; found: C, 75.69; H, 6.52; N, 3.39.

*N-cyclohexyl-N-(diphenylphosphanyl)-1,1-diphenylphosphanamine (L3):* After extracting in toluene, *L3* was recovered as a white solid with a yield of 82%. ^1^H NMR (400 MHz, CDCl_3_) δ 7.24 (m, 20H, Ar), 3.29–3.11 (m, 1H, cyclohexyl CH), 1.93–1.74 (m, 2H, cyclohexyl CH_2_), 1.64–1.30 (m, 4H, cyclohexyl CH_2_), 1.12–0.89 (m, 4H, cyclohexyl CH_2_). ^13^C{^1^H}-NMR (101 MHz, CDCl_3_) δ 140.00 (4C, Ar quaternary (PPh_2_)), 132.64 (8C, *o*-Ar CH (PPh_2_)), 128.48 (8C, *m*-Ar CH (PPh_2_)), 127.96 (4 C, *p*-Ar CH (PPh_2_)), 60.33 (1 C, cyclohexyl CH), 34.89 (1 C, cyclohexyl CH_2_), 26.12 (2C, cyclohexyl CH_2_), 25.47 (2C, cyclohexyl CH_2_). ^31^P NMR (162 MHz, CDCl_3_) δ 55.78. UV-vis: maximum absorbance at 276 nm. IR: 2927 cm^−1^ (C–H sp^3^ stretching), 1449–1433 cm^−1^ (two bands C=C aromatic stretching), 1091 cm^−1^ (C–P stretching), 1056 cm^−1^ (C–N stretching). Elemental analysis calcd (%) for C_30_H_31_NP_2_: C, 77.07; H, 6.68; N, 3.00; found: C, 76.84; H, 6.88; N, 3.31.

*N-(diphenylphosphanyl)-N,1,1-triphenylphosphanamine (L4):* After extracting in toluene, *L4* was recovered as a white solid with a yield of 91%. ^1^H NMR (400 MHz, CDCl_3_) δ 7.40–7.21 (m, 20H), 6.99–6.88 (m, 3H), 6.64 (d, *J* = 6.2 Hz, 2H). ^13^C{^1^H}-NMR (101 MHz, CDCl_3_) δ 159.32 (2C, *o*-Ar quaternary (*o*-Ph)), 158.02 (1C, Ar quaternary (*o*-Ph)), 140.28 (4C, Ar quaternary (PPh_2_)), 132.32 (8C, *o*-Ar CH (PPh_2_)), 128.03 (2C, *m*-Ar (*o*-Ph)), 127.65 (8C, *m*-Ar CH (PPh_2_)), 126.92 (4C, *p*-Ar CH (PPh_2_)), 122.03 (2C, *o*-Ar (*o*-Ph)), 117.22 (1C, *p*-Ar (*o*-Ph)). ^31^P NMR (162 MHz, CDCl_3_) δ 68.46. UV-vis: maximum absorbance at 280 nm. IR: 3057 cm^−1^ (C–H sp^2^ stretching), 1488–1431 cm^−1^ (two bands C=C aromatic stretching), 1172 cm^−1^ (C–N stretching), 1091 cm^−1^ (C–P stretching). Elemental analysis calcd (%) for C_30_H_25_NP_2_: C, 78.08; H, 5.46; N, 3.04; found: C, 78.12; H, 5.54; N, 3.30.

*N-(diphenylphosphanyl)-N-(2-fluorophenyl)-1,1-diphenylphosphanamine (L5):* After extracting in toluene, *L5* was recovered as a white solid with a yield of 80%. ^1^H NMR (400 MHz, CDCl_3_) δ 7.40–7.18 (m, 20 H, Ar (PPh_2_)), 7.04–6.96 (m, 1H*, o*-Ar (*o*-FPh)), 6.87–6.77 (m, 1H, *p*-Ar (*o*-FPh)), 6.76–6.62 (m, 2H, *m*-Ar (*o*-FPh)). ^13^C{^1^H}-NMR (101 MHz, CDCl_3_) δ 160.47 (1C, *o*-Ar quaternary (*o*-FPh)), 158.02 (1C, Ar quaternary (*o*-FPh)), 139.18 (4C, Ar quaternary (PPh_2_)), 133.39 (8C, *o*-Ar CH (PPh_2_)), 131.43 (1C, *m*-Ar (*o*-FPh)), 129.05 (8C, *m*-Ar CH (PPh_2_)), 127.97 (4C, *p*-Ar CH (PPh_2_)), 127.08 (1C, *o*-Ar (*o*-FPh)), 123.59 (1C, *m*-Ar (*o*-FPh)), 116.21 (1C, *p*-Ar (*o*-FPh)). UV-vis: maximum absorbance at 280 nm. IR: 3057 cm^−1^ (C–H sp^2^ stretching), 1479–1432 cm^−1^ (two bands C=C aromatic stretching), 1245 cm^−1^ (C–F stretching), 1181 cm^−1^ (C–N stretching), 1089 cm^−1^ (C–P stretching). Elemental analysis calcd (%) for C_30_H_24_FNP_2_: C, 75.15; H, 5.05; N, 2.92; found: C, 74.92; H, 5.11; N, 3.18.

*N-(diphenylphosphanyl)-N-(4-fluorophenyl)-1,1-diphenylphosphanamine (L6):* After extracting in toluene, *L6* was recovered as a white solid with a yield of 86%. ^1^H NMR (400 MHz, CDCl_3_) δ 7.38–7.19 (m, 20H, Ar), 6.58 (t, J = 8.7 Hz, 2H, *m-*Ar (*p*-FPh)), 6.48 (dd, J = 8.9, 5.1 Hz, 2H, *o-*Ar (*p*-FPh)). ^13^C{^1^H}-NMR (101 MHz, CDCl_3_) δ 161.44 (1C, *p*-Ar quaternary(*p*-FPh)), 159.01 (1C, Ar quaternary (*p*-FPh)), 138.93 (4C, Ar quaternary (PPh_2_)), 133.20 (8C, *o*-Ar CH (PPh_2_)), 130.67 (2C, *m-*Ar CH (*p*-FPh)), 129.11 (8C, *m*-Ar CH (PPh_2_)), 128.00 (4C, *p*-Ar CH (PPh_2_)), 114.79 (2C, *o-*Ar CH (*p*-FPh)). UV-vis: maximum absorbance at 278 nm. IR: 3056 cm^−1^ (C-H sp^2^ stretching), 1496–1430 cm^−1^ (two bands C=C aromatic stretching), 1202 cm^−1^ (C–F stretching), 1151 cm^−1^ (C–N stretching), 1091 cm^−1^ (C–P stretching). Elemental analysis calcd (%) for C_30_H_24_FNP_2_: C, 75.15; H, 5.05; N, 2.92; found: C, 75.33; H, 5.27; N, 3.11.

*N-(3-bromophenyl)-N-(diphenylphosphanyl)-1,1-diphenylphosphanamine (L7):* After extracting in toluene, *L7* was recovered as a white solid with a yield of 92%**.**
^1^H NMR (400 MHz, CDCl_3_) δ 7.40–7.19 (m, 20H, Ar (PPh_2_)), 7.08 (d, *J = 8.2 Hz*, 1H, *o-*Ar (*m-*BrPh)), 6.81 (t, *J = 8.0 Hz*, 1H, *m-*Ar (*m-*BrPh)), 6.68 (s, 1H, *o’-*Ar (*m-*BrPh)), 6.62 (d, *J = 8.0 Hz*, 1H, *p-*Ar (*m-*BrPh)). ^13^C{^1^H}-NMR (101 MHz, CDCl_3_) δ 148.66 (4C, Ar quaternary (*m-*BrPh)), 138.57 (2C, *m*’-Ar quaternary (*m-*BrPh) and Ar quaternary (PPh_2_)), 133.13 (8C, *o*-Ar CH (PPh_2_)), 131.95 (1C, *p-*Ar (*m-*BrPh)), 129.20 (8C, *m*-Ar CH (PPh_2_)), 128.04 (4C, *p*-Ar CH (PPh_2_)), 127.43 (2C, *o-*Ar (*m-*BrPh)), 121.36 (1C, *m-*Ar (*m-*BrPh)). ^31^P NMR (162 MHz, CDCl_3_) δ 68.65. UV-vis: maximum absorbance at 282 nm. IR: 3072 cm^−1^ (C–H sp^2^ stretching), 1465–1432 cm^−1^ (two bands C=C aromatic stretching), 1203 cm^−1^ (C–N stretching), 1091 cm^−1^ (C–P stretching), 690 cm^−1^ (C–Br stretching). Elemental analysis calcd (%) for C_30_H_24_BrNP_2_: C, 66.68; H, 4.48; N, 2.59; found: C, 66.92; H, 4.53; N, 2.75.

*N-(4-chlorophenyl)-N-(diphenylphosphanyl)-1,1-diphenylphosphanamine (L8):* After extracting in toluene, *L8* was recovered as a white solid with a yield of 90%. ^1^H NMR (400 MHz, CDCl_3_) δ 7.36–7.19 (m, 20H, Ar (PPh_2_)),), 6.85 (d, *J = 8.7 Hz*, 2H, *m-*Ar CH (*p-*ClPh)), 6.48 (d, J = 8.8 Hz, 2H, *o-*Ar CH(*p-*ClPh)). ^13^C{^1^H}-NMR (101 MHz, CDCl_3_) δ 145.62 (1C, Ar quaternary (*p-*ClPh)), 138.72 (5C, *p*-Ar quaternary(*p-*ClPh) and Ar quaternary (PPh_2_)), 133.15 (8C, *o*-Ar CH (PPh_2_)), 130.22 (2C, *m-*Ar CH (*p-*ClPh)), 129.15 (8C, *m*-Ar CH (PPh_2_)), 128.17 (2C, *o-*Ar CH (*p-*ClPh)), 128.06 (4C, *p*-Ar CH (PPh_2_)). ^31^P NMR (162 MHz, cdcl_3_) δ 69.64. UV-vis: maximum absorbance at 282 nm. IR: 3036 cm^−1^ (C–H sp^2^ stretching), 1477–1431 cm^−1^ (two bands C=C aromatic stretching), 1211 cm^−1^ (C–N stretching), 1092 cm^−1^ (C–P stretching), 901 cm^−1^ (C–Cl stretching). Elemental analysis calcd (%) for C_30_H_24_ClNP_2_: C, 72.66; H, 4.88; N, 2.82; found: C, 72.45; H, 4.46; N, 2.62.

*N,N-bis(diphenylphosphanyl)quinolin-6-amine (L9):* After extracting in toluene, *L9* was recovered as a yellow solid with a yield of 78%. ^1^H NMR (400 MHz, CDCl_3_) δ 8.73 (dd, *J* = 4.2, 1.7 Hz, 1H, Ar CH8 (quinoline)), 7.67 (t, *J* = 9.6 Hz, 2H, Ar CH9 and Ar CH10 (quinoline)), 7.41–7.32 (m, 8H, *o-*Ar (PPh_2_)), 7.33–7.23 (m, 12H, *m-*Ar and *p-*Ar (PPh_2_)), 7.20 (dd, *J* = 8.3, 4.3 Hz, 1H, Ar CH3 (quinoline)), 7.10 (d, *J* = 9.0 Hz, 1H, Ar CH2 (quinoline)), 6.93 (s, 1H, Ar CH6 (quinoline)). ^13^C{^1^H}-NMR (101 MHz, CDCl_3_) δ 149.65 (1 C, C8 (quinoline)), 145.95 (1 C, C1 (quinoline)), 145.66 (1 C, C4 (quinoline)), 138.75 (4 C, Ar quaternary (PPh_2_)), 135.66 (1 C, C10 (quinoline)), 133.17 (8 C, *o*-Ar CH (PPh_2_)), 131.77 (1 C, C5 (quinoline)), 129.19 (8 C, *m*-Ar CH (PPh_2_)), 129.12 (1 C, C2 (quinoline)), 128.10 (1 C, C3 (quinoline)), 127.94 (4 C, *p*-Ar CH (PPh_2_)), 125.90 (1 C, C6 (quinoline)), 120.89 (1 C, C9 (quinoline)). ^31^P NMR (162 MHz, cdcl_3_) δ 68.73. UV-vis: maximum absorbance at 282 nm. IR: 3047 cm^−1^ (C–H sp^2^ stretching), 1492–1431 cm^−1^ (two bands C=C aromatic stretching), 1210 cm^−1^ (C–N stretching), 1094 cm^−1^ (C–P stretching). Elemental analysis calcd (%) for C_33_H_26_N_2_P: C, 77.33; H, 5.11; N, 5.47; found: C, 77.22; H, 5.32; N, 5.81.

*(E)-N-(diphenylphosphanyl)-1,1-diphenyl-N-(4-(phenyldiazenyl)phenyl)phosphanamine (L10):* After extracting in toluene, *L10* was recovered as a yellow solid with a yield of 79%**.**
^1^H NMR (400 MHz, CDCl_3_) δ 7.82 (d, *J* = 6.9 Hz, 2H, H3 and H5 (azobenzene)), 7.53 (d, *J* = 8.8 Hz, 2H, H10 and H14 (azobenzene)), 7.45 (d, *J* = 7.5 Hz, 2H, H11 and H13 (azobenzene)), 7.42–7.38 (m, 8H, *o-*Ar (PPh_2_)), 7.34 (t, *J* = 1.5 Hz, 1H, H12 (azobenzene)), 7.33–7.23 (m, 12H, *m-*Ar and *p-*Ar (azobenzene)), 6.81 (d, *J* = 8.8 Hz, 2H, H2 and H6 (azobenzene)). ^13^C{^1^H}-NMR (101 MHz, CDCl_3_) δ 152.59 (1 C, C9 (azobenzene)), 150.47 (1 C, C1 (azobenzene)), 149.38 (1 C, C4 (azobenzene)), 138.55 (4 C, Ar quaternary (PPh_2_)), 133.15 (8 C, *o*-Ar CH (PPh_2_)), 130.72 (2 C, C3 and C5 (azobenzene)), 129.19 (8 C, *m*-Ar CH (PPh_2_)), 129.02 (2 C, C11 and C13 (azobenzene)), 128.83 (2 C, C10 and C14 (azobenzene)), 128.12 (4 C, *p*-Ar CH (PPh_2_)), 122.82 (1 C, C2 (azobenzene)), 122.66 (1 C, C6 (azobenzene)). UV-vis: maximum absorbance at 358 nm. IR: 3047 cm^−1^ (C–H sp^2^ stretching), 1492 cm^−1^ (N=N stretching), 1478–1434 cm^−1^ (two bands C=C aromatic stretching), 1214 cm^−1^ (C–N stretching), 1091 cm^−1^ (C–P stretching). Elemental analysis calcd (%) for C_36_H_29_N_3_P_2_: C, 76.45; H, 5.17; N, 7.43; found: C, 76.71; H, 5.32; N, 7.75.

### 2.3. Synthesis and Characterization of Ruthenium Compounds Ru1–Ru10

To a mixture of ligand (0.07 mmol), [RuCl_2_(p-cym)]_2_ (0.023 g, 0.037 mmol) and NaBF_4_ (0.010 g, 0.091 mmol) was added methanol (3 mL) and the mixture was allowed to stir for 4 h at room temperature. After this time, a white precipitate appeared. The complexes were isolated as orange solids after filtration.

*[RuCl(p-cym)({Ph_2_P}_2_N{C_4_H_9_})][BF_4_] (Ru1).* Yield: 46.32 mg, 0.064 mmol, 85%. ^1^H NMR (400 MHz, CDCl_3_) δ 8.23–8.10 (d, J = 6.6 Hz, 8 H, *o*-Ar), 7.85 (t, *J* = 7.4 Hz, 4 H, *p*-Ar), 7.67 (t, *J* = 7.3 Hz, 8 H, *m*-Ar), 5.88 (d, *J* = 6.4 Hz, 2 H, cym), 5.85 (d, *J* = 6.5 Hz, 2 H, cym), 2.60–2.49 (m, 1 H, CH_3_–**CH**–CH_3_), 1.95 (d, *J* = 5.9 Hz, 3 H, CH_3_), 0.96 (d, *J* = 6.9 Hz, 6 H, **CH_3_**–CH–**CH_3_**), 0.71 (s, 9 H, CH_3_). ^13^C{^1^H}-NMR (101 MHz, CDCl_3_) δ 135.36 (8 C, *o*-Ar CH (PPh_2_), 132.64 (8 C, *m*-Ar CH (PPh_2_), 130.22 (4 C, *p*-Ar CH (PPh_2_), 98.50 (4 C, cym), 31.82 (1 C, **CH**–CH_3_)_2_), 31.18 (3 C, CH_3_), 22.81 (2 C, **CH_3_**–CH), 18.62 (1 C, CH_3_). Quaternary carbons were not found. ^31^P NMR (162 MHz, CDCl_3_) δ 79.91. UV-vis: maximum absorbance at 280 nm. IR: 2963 cm^−1^ (C–H sp^3^ stretching), 1474–1436 cm^−1^ (two bands C=C aromatic stretching), 1090 cm^−1^ (C–N stretching), 1053 cm^−1^ (C–P stretching), 701 cm^−1^ (Ru–Cl stretching). Elemental analysis calcd (%) for C_38_H_43_BClF_4_NP_2_Ru: C, 57.12; H, 5.42; N, 1.75; found: C, 57.93; H, 5.15; N, 1.97. MS (ESI) m/z for [C_38_H_43_ClNP_2_Ru^+^]: 712.16 (100%).

*[RuCl(p-cym)({Ph_2_P}_2_N{C_3_H_7_})][BF_4_] (Ru2).* Yield: 49.41 mg, 0.070 mmol, 93%. ^1^H NMR (400 MHz, CDCl_3_) δ 7.54 (dd, *J* = 61.7, 29.3 Hz, 20 H, Ar), 6.05 (s, 2 H, cym), 5.79 (s, 2 H, cym), 2.75 (t, *J =* 4.3 Hz*,* 2 H, **CH_2_**–CH_2_–CH_3_), 2.55 (s, 1 H, CH_3_–**CH**–CH_3_), 1.79 (s, 3 H, CH_3_), 1.10–0.94 (m, 2 H, CH_2_–**CH_2_**–CH_3_), 1.06 (d, *J* = 6.8 Hz, 6 H, **CH_3_**–CH–**CH_3_**), 0.40 (s, 3 H, CH_2_–CH_2_–**CH_3_**). ^13^C{^1^H}-NMR (101 MHz, CDCl_3_) δ 133.55 (4 C, Ar quaternary (PPh_2_)), 132.65 (8 C, *m*-Ar CH (PPh_2_)), 130.07 (8 C, *o*-Ar CH (PPh_2_)), 127.95 (4 C, *p*-Ar CH (PPh_2_)), 117.87 (1 C, cym quaternay), 111.04 (1 C, cym quaternary) 95.95 (2 C, cym), 90.19 (2 C, cym), 52.26 (1 C, **CH_2_**–CH_2_–CH_3_), 31.77 (1 C, **CH**–CH_3_)_2_), 23.46 (1 C, CH_2_–**CH_2_**–CH_3_), 22.79 (2 C, **CH_3_**–CH), 18.85 (1 C, CH_3_), 11.15 (1 C, CH_2_–CH_2_–**CH_3_**). ^31^P NMR (162 MHz, CDCl_3_) δ 76.56. UV-vis: maximum absorbance at 276 nm. IR: 2966 cm^−1^ (C–H sp^3^ stretching), 1477–1435 cm^−1^ (two bands C=C aromatic stretching), 1091 cm^−1^ (C–N stretching), 1050 cm^−1^ (C–P stretching), 695 cm^−1^ (Ru–Cl stretching). Elemental analysis calcd (%) for C_37_H_41_BClF_4_NP_2_Ru: C, 56.61; H, 5.26; N, 1.78; found: C, 56.68; H, 5.48; N, 1.87. MS (ESI) m/z for [C_37_H_41_ClNP_2_Ru^+^]: 698.14 (100%).

*[RuCl(p-cym)({Ph_2_P}_2_N{C_6_H_11_})][BF_4_] (Ru3).* Yield: 53.11 mg, 0.072 mmol, 92%. ^1^H NMR (400 MHz, CDCl_3_) δ 7.89–7.42 (m, 20 H, Ar), 5.99 (t, *J* = 6.8 Hz, 2 H, cym), 5.95 (d, *J* = 6.2 Hz, 2 H, cym), 2.66–2.53 (m, 1 H, CH_3_–**CH**–CH_3_), 2.03 (s, 3 H, CH_3_), 1.23 (t, *J* = 16.6 Hz, 4 H, cyclohexyl CH_2_), 1.11 (d, *J* = 11.5 Hz, 2 H, cyclohexyl CH_2_), 1.02 (d, *J* = 6.9 Hz, 6 H, **CH_3_**–CH–**CH_3_**), 0.63 (t, *J* = 13.1 Hz, 1 H, cyclohexyl CH), 0.55–0.40 (m, 4 H, cyclohexyl CH_2_). ^13^C{^1^H}-NMR (101 MHz, CDCl_3_) δ 134.33 (8 C, *o*-Ar CH (PPh_2_)), 133.71 (4 C, Ar quaternay (PPh_2_)) 132.36 (8 C, *m*-Ar CH (PPh_2_)), 130.03 (4 C, *p*-Ar CH (PPh_2_)), 116.76 (1 C, cym quaternary), 112.54 (1 C, cym quaternary), 96.29 (2 C, cym), 89.63 (2 C, cym), 33.19–32.77 (2 C, cyclohexyl CH_2_), 31.73 (1 C, **CH**–CH_3_)_2_), 25.95 (2 C, cyclohexyl CH_2_), 24.51 (2 C, cyclohexyl CH_2_ and cyclohexyl CH), 22.83 (2 C, **CH_3_**–CH), 19.01 (1 C, CH_3_). ^31^P NMR (162 MHz, CDCl_3_) δ 75.97. UV-vis: maximum absorbance at 276 nm. IR: 2941 cm^−1^ (C–H sp^3^ stretching), 1478–1434 cm^−1^ (two bands C=C aromatic stretching), 1075 cm^−1^ (C–N stretching), 1046 cm^−1^ (C–P stretching), 699 cm^−1^ (Ru–Cl stretching). Elemental analysis calcd (%) for C_40_H_45_BClF_4_NP_2_Ru: C, 58.23; H, 5.50; N, 1.70; found: C, 58.42; H, 5.39; N, 1.61. MS (ESI) m/z for [C_40_H_45_ClNP_2_Ru^+^]: 731.18 (100%).

*[RuCl(p-cym)({Ph_2_P}_2_N{C_6_H_5_})][BF_4_] (Ru4).* Yield^:^ 51.31 mg, 0.070 mmol, 92%.^1^H NMR (400 MHz, CDC_3_) δ 7.68–7.56 (m, 8 H, *o*-Ar (PPh_2_)), 7.54–7.43 (m, 8 H, *m*-Ar (PPh_2_)), 7.35 (t, *J* = 7.6 Hz, 4 H, *p*-Ar (PPh_2_)), 7.04 (t, *J* = 7.3 Hz, 1 H, *p*-phenyl), 6.96 (t, *J* = 7.7 Hz, 2 H, *m*-phenyl), 6.54 (d, 2 H, *o*-phenyl), 6.06 (d, *J* = 6.3 Hz, 2 H, cym), 5.83 (d, *J* = 6.3 Hz, 2 H, cym), 2.57–2.47 (m, *J* = 13.8, 6.8 Hz, 1 H, CH_3_–**CH**–CH_3_), 1.68 (s, 3 H, CH_3_), 1.09 (d, *J* = 6.9 Hz, 6 H, **CH_3_**–CH–**CH_3_**). ^13^C{^1^H}-NMR (101 MHz, CDCl_3_) δ 139.66 (1 C, phenyl quaternary), 133.41 (8 C, *o*-Ar CH (PPh_2_)), 132.79 (4 C, Ar quaternary (PPh_2_), 129.82 (8 C, *m*-Ar CH (PPh_2_)), 128.97 (2 C, phenyl *m*-Ar CH), 127.83 (4 C, *p*-Ar CH (PPh_2_)), 126.73 (1 C, phenyl *p*-Ar CH), 126.51 (2 C, phenyl *o*-Ar CH), 120.58 (1 C, cym quaternary), 109.01 (1 C, cym quaternary), 94.69–93.57 (2 C, cym), 92.19 (2 C, cym), 31.44 (1 C, **CH**-(CH_3_)_2_), 22.49 (2 C, **CH_3_**-CH), 18.00 (1 C, CH_3_). ^31^P NMR (162 MHz, CDCl_3_) δ 79.01. UV-vis: maximum absorbance at 276 nm. IR: 2962 cm^−1^ (C-H sp^3^ stretching), 1482–1435 cm^−1^ (two bands C=C aromatic stretching), 1089 cm^−1^ (C–N stretching), 1031 cm^−1^ (C–P stretching), 691 cm^−1^ (Ru–Cl stretching). Elemental analysis calcd (%) for C_40_H_39_BClF_4_NP_2_Ru: C, 58.66; H, 4.80; N, 1.71; found: C, 58.79; H, 4.48; N, 1.97. MS (ESI) m/z for [C_40_H_38_ClFNP_2_Ru^+^]: 731.18 (100%).

*[RuCl(p-cym)({Ph_2_P}_2_N{C_6_H_4_F})][BF_4_] (Ru5).* Yield: 50.19 mg, 0.067 mmol, 89%. ^1^H NMR (400 MHz, CDCl_3_) δ 7.57 (s, 8 H, *o*-Ar (PPh_2_)), 7.50–7.39 (m, 8 H, *m*-Ar (PPh_2_)), 7.29 (t, *J* = 7.5 Hz, 4 H, *p*-Ar (PPh_2_)), 7.14 (t, *J* = 10.2 Hz, 1 H, *p*-Ar (*o*-FPh)), 6.97 (t, *J* = 9.1 Hz, 1 H, *m’*-Ar (*o*-FPh)), 6.62 (t, *J* = 7.6 Hz, 1 H, *m*-Ar (*o*-FPh)), 6.17 (d, *J* = 6.2 Hz, 2 H, cym), 6.08 (t, *J* = 7.6 Hz, 1 H, *o*-Ar (*o*-FPh)), 5.90 (d, *J* = 6.0 Hz, 2 H, cym), 2.60–2.47 (m, 1 H, CH_3_-**CH**-CH_3_), 1.69 (s, 3 H, CH_3_), 1.11 (d, *J* = 6.9 Hz, 6 H, **CH_3_**-CH-**CH_3_**). ^13^C{^1^H}-NMR (101 MHz, CDCl_3_) δ 134.15 (1 C, *o’*-Ar quaternary (*o*-FPh), 134.00 (1 C, *o*-Ar (*o*-FPh)), 132.94 (8 C, *m*-Ar CH (PPh_2_)), 131.94 (4 C, Ar quaternary (PPh_2_)), 130.53 (1 C, *p*-Ar (*o*-FPh)), 129.43 (8 C, *o*-Ar CH (PPh_2_)), 128.12 (1 C, Ar quaternary (*o*-FPh)), 127.69 (4 C, *p*-Ar CH (PPh_2_)), 124.25 (1 C, *m*-Ar (*o*-FPh)), 120.59 (1 C, cym quaternary), 116.96 (1 C, *m’*-Ar (*o*-FPh)), 108.09 (1 C, cym quaternary), 93.89 (2 C, cym), 92.57 (2 C, cym), 31.31 (1 C, **CH**–(CH_3_)_2_), 22.47 (2 C, **CH_3_**–CH), 17.99–16.94 (1 C, CH_3_). ^31^P NMR (162 MHz, CDCl_3_) δ 84.21. UV-vis: maximum absorbance at 272 nm. IR: 3065 cm^−1^ (C–H sp^2^ stretching), 2968 cm^−1^ (C–H sp^3^ stretching), 1485–1435 cm^−1^ (two bands C=C aromatic stretching), 1257 cm^−1^ (C–F stretching), 1097 cm^−1^ (C–N stretching), 1050 cm^−1^ (C–P stretching), 691 cm^−1^ (Ru–Cl stretching). Elemental analysis calcd (%) for C_40_H_38_BClF_5_NP_2_Ru: C, 57.40; H, 4.58; N, 1.67; found: C, 57.72; H, 4.32; N, 1.83. MS (ESI) m/z for [C_40_H_38_ClFNP_2_Ru^+^]: 750.12 (100%).

*[RuCl(p-cym)({Ph_2_P}_2_N{C_6_H_4_F})][BF_4_] (Ru6).* Yield: 51.40 mg, 0.068 mmol, 90%. ^1^H NMR (400 MHz, CDCl_3_) δ 7.67–7.32 (m, 20 H, Ar (PPh_2_)), 6.71–6.63 (t, 2 H, *o*-Ar (*p*-FPh)), 6.52–6.45 (m, 2 H, *m*-Ar (*p*-FPh)), 6.15 (d, *J* = 6.3 Hz, 2 H, cym), 5.90 (d, *J* = 6.2 Hz, 2 H, cym), 2.60–2.50 (m, 1 H, CH_3_–**CH**–CH_3_), 1.74 (s, 3 H, CH_3_), 1.09 (d, *J* = 6.9 Hz, 6 H, **CH_3_**-CH-**CH_3_**). ^13^C{^1^H}-NMR (101 MHz, CDCl_3_) δ 133.62 (1 C, *p*-Ar quaternary (*p*-FPh)), 133.38 (4 C, Ar quaternary (PPh_2_)), 133.22 (8 C, Ar CH (PPh_2_)), 131.94 (1 C, Ar quaternary (*p*-FPh)), 130.11 (2 C, *m*-Ar CH (*p*-FPh)), 129.83 (8 C, Ar CH (PPh_2_)), 127.81 (4 C, Ar CH (PPh_2_)), 119.96 (1 C, cym quaternary), 115.83 (2 C, *o*-Ar CH (*p*-FPh)), 110.03 (1 C, cym quaternary), 94.94 (2 C, cym), 91.74 (2 C, cym), 31.59 (1 C, **CH**–(CH_3_)_2_), 22.61 (2 C, **CH_3_**–CH), 18.26 (1 C, CH_3_). ^31^P NMR (162 MHz, CDCl_3_) δ 81.02. UV-vis: maximum absorbance at 274 nm. IR: 3071 cm^−1^ (C–H sp^2^ stretching), 2965 cm^−1^ (C–H sp^3^ stretching), 1503–1434 cm^−1^ (two bands C=C aromatic stretching), 1212 cm^−1^ (C–F stretching), 1093 cm^−1^ (C–N stretching), 1050 cm^−1^ (C–P stretching), 691 cm^−1^ (Ru–Cl stretching). Elemental analysis calcd (%) for C_40_H_38_BClF_5_NP_2_Ru: C, 57.40; H, 4.58; N, 1.67; found: C, 57.51; H, 4.61; N, 1.88. MS (ESI) m/z for [C_40_H_38_BrClNP_2_Ru^+^]: 810.04 (100%).

*[RuCl(p-cym)({Ph_2_P}_2_N{C_6_H_4_Br})][BF_4_] (Ru7)*. Yield: 56.04 mg, 0.069 mmol, 92%. ^1^H NMR (400 MHz, CDCl_3_) δ 7.74–7.33 (m, *J* = 51.1, 40.4, 17.9 Hz, 22 H, Ar (PPh_2_), *o*-Ar CH (*m*-BrPh) and *m*-Ar CH (*m*-BrPh)), 6.93 (d, *J* = 8.7 Hz, 1 H, *p*-Ar CH (*m*-BrPh)), 6.44 (d, *J* = 8.7 Hz, 1 H, *o*-Ar CH (*m*-BrPh)), 6.12 (d, 2 H, cym), 5.91 (d, 2 H, cym), 2.59–2.49 (m, 1 H, CH_3_–**CH**–CH_3_), 1.74 (s, 3 H, CH_3_), 1.09 (d, *J* = 6.6 Hz, 6 H, **CH_3_**–CH–**CH_3_**). ^13^C{^1^H}-NMR (101 MHz, CDCl_3_) δ 133.66 (1 C, *m’*-Ar quaternary (*m*-BrPh)), 133.26 (4 C, Ar quaternary (PPh_2_)), 132.04 (8 C, Ar CH (PPh_2_)), 130.09 (1 C, Ar quaternary (*m*-BrPh)), 129.81 (1 C, *m*-Ar CH (*m*-BrPh)), 129.57 (8 C, Ar CH (PPh_2_)), 128.02 (4 C, Ar CH (PPh_2_)), 127.61 (1 C, *o’*-Ar CH (*m*-BrPh)), 124.81 (1 C, *o*-Ar CH (*m*-BrPh)), 122.31 (1 C, *p*-Ar CH (*m*-BrPh)), 110.18 (1 C, cym quaternary), 109.26 (1 C, cym quaternary), 95.47 (2 C, cym), 92.17 (2 C, cym), 31.62 (1 C, **CH**–(CH_3_)_2_), 22.64 (2 C, **CH_3_**–CH), 18.35 (1 C, CH_3_). ^31^P NMR (162 MHz, CDCl_3_) δ 81.21. UV-vis: maximum absorbance at 257 nm. IR: 3053 cm^−1^ (C-H sp^2^ stretching), 2968 cm^−1^ (C–H sp^3^ stretching), 1475–1435 cm^−1^ (two bands C=C aromatic stretching), 1092 cm^−1^ (C–N stretching), 1050 cm^−1^ (C–P stretching), 700 cm^−1^ (Ru–Cl stretching). Elemental analysis calcd (%) for C_40_H_38_BBrClF_4_NP_2_Ru: C, 53.51; H, 4.27; N, 1.56; found: C, 54.19; H, 4.59; N, 1.34. MS (ESI) m/z for [C_40_H_38_BrClNP_2_Ru^+^]: 810.04 (100%).

*[RuCl(p-cym)({Ph_2_P}_2_N{C_6_H_4_Cl})][BF_4_] (Ru8).* Yield: 51.09 mg, 0.066 mmol, 88%. ^1^H NMR (400 MHz, CDCl_3_) δ 7.66 (d, *J* = 29.2 Hz, 8 H, *o*-Ar (PPh_2_)), 7.49 (t, *J* = 8.1, 8 H, *m*-Ar (PPh_2_)), 7.39 (t, *J* = 6.8 Hz, 4 H, *p*-Ar (PPh_2_)), 6.93 (d, *J* = 8.6 Hz, 2 H, *o*-Ar (*p*-ClPh)), 6.45 (d, *J* = 8.6 Hz, 2 H, *m*-Ar (*p*-ClPh)), 6.12 (d, 2 H, cym), 5.90 (d, 2 H, cym, 2.59–2.49 (m, 1 H, CH_3_–**CH**–CH_3_), 1.75 (s, 3 H, CH_3_), 1.09 (d, *J* = 5.7 Hz, 6 H, **CH_3_**–CH–**CH_3_**). ^13^C{^1^H}-NMR (101 MHz, CDCl_3_) δ 138.13 (1 C, *p*-Ar quaternary (*p*-ClPh)), 133.53 (1 C, Ar quaternary (*p*-ClPh)), 133.40 (8 C, *o*-Ar CH (PPh_2_)), 133.17 (8 C, *m*-Ar CH (PPh_2_)), 132.01 (4 C, Ar quaternary (PPh_2_)), 129.18 (2 C, Ar CH (*p*-ClPh)), 128.10 (2 C, Ar CH (*p*-ClPh)), 127.98 (4 C, *p*-Ar CH (PPh_2_)), 120.44 (1 C, cym quaternary), 109.75 (1 C, cym quaternary), 95.20 (2 C, cym), 92.12 (2 C, cym), 31.58 (1 C, **CH**–(CH_3_)_2_), 22.60 (2 C, **CH_3_**–CH), 18.23 (1 C, CH_3_). ^31^P NMR (162 MHz, CDCl_3_) δ 80.65. UV-vis: maximum absorbance at 272 nm. IR: 3065 cm^−1^ (C–H sp^2^ stretching), 2969 cm^−1^ (C–H sp^3^ stretching), 1474–1434 cm^−1^ (two bands C=C aromatic stretching), 1094 cm^−1^ (C–N stretching), 1051 cm^−1^ (C–P stretching), 890 cm^−1^ (C–Cl stretching), 696 cm^−1^ (Ru–Cl stretching). Elemental analysis calcd (%) for C_40_H_38_BCl_2_F_4_NP_2_Ru: C, 56.29; H, 4.49; N, 1.64; found: C, 56.45; H, 4.26; N, 1.98. MS (ESI) m/z for [C_40_H_38_Cl_2_NP_2_Ru^+^]: 766.09 (100%).

*[RuCl(p-cym)({Ph_2_P}_2_N{C_9_H_6_N})][BF_4_] (Ru9).* Yield**:** 53.20 mg, 0.068 mmol, 90%. ^1^H NMR (400 MHz, CDCl_3_) δ 7.76–7.32 (m, 24 H, Ar (PPh_2_) and Ar CH3, CH8, CH9 and CH10 (quinoline)), 7.03–6.98 (m, 1 H, Ar CH2 (quinoline)), 6.76 (d, *J* = 1.5 Hz, 1 H, Ar CH6 (quinoline)), 6.12 (d, *J* = 6.2 Hz, 2 H, cym), 5.90 (d, *J* = 6.5 Hz, 2 H, cym), 2.61–2.52 (m, 1 H, CH_3_–**CH**–CH_3_), 1.76 (s, 3 H, CH_3_), 1.12 (d, *J* = 6.9 Hz, 6 H, **CH_3_**–CH–**CH_3_**). ^13^C{^1^H}-NMR (101 MHz, CDCl_3_) δ 133.51 (1 C, C8 (quinoline)), 133.36 (8 C, *o*-Ar CH (PPh_2_)), 133.10 (1 C, C10 (quinoline)), 132.00 (8 C, *m*-Ar CH (PPh_2_)), 130.01 (1 C, C3 (quinoline)), 128.18 (1 C, C2 (quinoline)), 127.98 (4 C, *p*-Ar CH (PPh_2_)), 125.00 (1 C, C6 (quinoline)), 122.00 (1 C, C9 (quinoline)), 95.07 (2 C, cym), 92.19 (2 C, cym), 31.55 (1 C, **CH**–(CH_3_)_2_), 22.58 (2 C, **CH_3_**–CH), 18.20 (1 C, CH_3_). Quaternary carbons were not found. ^31^P NMR (162 MHz, CDCl_3_) δ 80.87. UV-vis: maximum absorbance at 257 nm. IR: 3057 cm^−1^ (C-H sp^2^ stretching), 2954 cm^−1^ (C-H sp^3^ stretching), 1498–1480 cm^−1^ (two bands C=C aromatic stretching), 1093 cm^−1^ (C–N stretching), 1046 cm^−1^ (C–P stretching), 693 cm^−1^ (Ru–Cl stretching). Elemental analysis calcd (%) for C_43_H_40_BClF_4_N_2_P_2_Ru: C, 59.36; H, 4.63; N, 3.22; found: C, 59.39; H, 4.66; N, 3.00. MS (ESI) m/z for [C_43_H_40_ClN_2_P_2_Ru^+^]: 783.14 (100%).

*[RuCl(p-cym)({Ph_2_P}_2_N{C_12_H_9_N_2_})][BF_4_] (Ru10).* Yield: 57.18 mg, 0.068 mmol, 91%. ^1^H NMR (400 MHz, CDCl_3_) δ 8.84–8.30 (m, *J* = 35.6, 14.9 Hz, 27 H, Ar (PPh_2_) and Ar CH3, CH5, CH10, CH11, CH12, CH13 and CH14 (azobencene)), 7.71 (d, *J* = 8.0 Hz, 2 H, Ar CH2 and CH6 (azobencene)), 7.08 (s, 2 H, cym), 6.88 (s, 2 H, cym), 3.65–3.45 (m, 1 H, **CH**–(CH_3_)_2_), 2.69 (s, 3 H, CH_3_), 2.13 (d, *J* = 6.8 Hz, 6 H, **CH_3_**–CH–**CH_3_**). ^13^C{^1^H}-NMR (101 MHz, CDCl_3_) δ 147.44 (1 C, C9 (quaternary azobencene)), 134.38 (1 C, C1 (quaternary azobencene)), 133.89 (1 C, Ar quaternary (PPh_2_)), 132.76 (8 C, Ar CH (PPh_2_)), 132.20 (1 C, C4 (quaternary azobencene)), 130.96 (8 C, Ar CH (PPh_2_)), 129.94 (5 C, *p*-Ar CH (PPh_2_) and C12 (azobencene)), 128.91 (2 C, C3 and C5 (azobencene)), 126.22 (2 C, C11 and C13 (azobencene)), 124.18 (2 C, C10 and C14 (azobencene)), 123.60 (2 C, C2 and C6 (azobencene)), 120.87 (1 C, cym quaternary), 109.99 (1 C, cym quaternary), 96.12 (2 C, cym), 93.32 (2 C, cym), 32.28 (1 C, **CH**–(CH_3_)_2_), 23.31 (2 C, **CH_3_**–CH), 18.84 (1 C, CH_3_). ^31^P NMR (162 MHz, CDCl_3_) δ 80.39. UV-vis: maximum absorbance at 348 nm. IR: 3065 cm^−1^ (C–H sp^2^ stretching), 2944 cm^−1^ (C–H sp^3^ stretching), 1493 cm^−1^ (N=N stretching), 1482–1435 cm^−1^ (two bands C=C aromatic stretching), 1093 cm^−1^ (C–N stretching), 1042 cm^−1^ (C–P stretching), 690 cm^−1^ (Ru–Cl stretching). Elemental analysis calcd (%) for C_46_H_43_BClF_4_N_3_P_2_Ru: C, 59.85; H, 4.70; N, 4.55; found: C, 60.12; H, 4.53; N, 4.42. MS (ESI) m/z for [C_46_H_43_ClN_3_P_2_Ru^+^]: 836.17 (100%).

### 2.4. X-ray Crystallography

Prismatic crystal for *Ru3* was mounted on a glass fiber and used for data collection on a Bruker D8 Venture with a Photon detector equipped with graphite monochromated *MoKα* radiation (λ = 0.71073 Å) (Bruker Apex2, Bruker AXS Inc., Madison, WI, USA, 2004). The data reduction was performed with the APEX2 software and corrected for absorption using SADABS [38]. Crystal structures were solved by direct methods using the SIR97 program [39] and refined by full-matrix least-squares on *F^2^* including all reflections using anisotropic displacement parameters by means of the WINGX crystallographic package [40,41]. Generally, anisotropic temperature factors were assigned to all atoms except for C2 and hydrogen atoms, which are riding their parent atoms with an isotropic temperature factor arbitrarily chosen as 1.2 times that of the respective parent. Several crystals of *Ru3* were measured and the structure was solved from the best data we were able to collect, due to the fact that the crystals diffracted very little. Final *R*(*F*), *wR*(*F*^2^), and goodness of fit agreement factors, details on the data collection and analysis can be found in Appendix A. CCDC 2,076,579 contain the supplementary crystallographic data for *Ru3.* These data can be obtained free of charge from The Cambridge Crystallographic Data Centre via www.ccdc.cam.ac.uk/data_request/cif, accessed on 11 April 2021.

### 2.5. HSA Binding Studies by Steady-State Fluorescence Spectroscopy

Three milliliters of protein solution (5 μM) was titrated by successive addition of a metallodrug stock solution. The excitation wavelength was 283 nm HSA, whereas the emission fluorescence intensities were collected at 372 nm. The excitation and emission slits were fixed at 5 nm. The step and dwell time was 0.1 s. For the inner filter effect fluorescence was corrected through:$$F_{corr}=F_{obs}\;10^{\frac{\left(A_{exc}+A_{em}\right)}{2}}$$
where *F_corr_* and *F_obs_* are the corrected and observed fluorescence intensities, and *A_exc_* and *A_em_* are the absorbances of the system at excitation and emission wavelengths. Temperature was controlled at 300 K by a temperature-controlled cuvette holder, TC 125 (Quantum Northwest), and the experiment was repeated three times.

The Stern–Volmer equation was then used to evaluate the quenching constant $K_{SV}$ of the studied drug–protein systems:$$\frac{F_{0}}{F}=1+K_{SV}\left[Q\right]$$
where $F_{0}$ and F are the emission intensities in the absence and the presence of the quencher, respectively, [Q] is the concentration of the quencher (metallodrug), and $K_{SV}$ is the Stern–Volmer constant [42].

For static quenching, the relationship between fluorescence intensity and the quencher concentration, [*Q*], is described by the following equation:$$\log\left(\frac{F_{0}-F}{F}\right)=\log K_{a}+n\;\log\left[Q\right]$$
where *F_0_* corresponds to the fluorescence intensity of the protein in absence of quencher, *n* and *K_a_* are the number of binding sites and binding constant, respectively [43,44].

### 2.6. Biological Assays

The compounds were dissolved in dimethyl sulfoxide (DMSO) before performing each experiment. The maximal concentration used was 50 μM, due to limited water solubility; cisplatin was tested up to 100 μM [45]. The same volume of solvent was added to control conditions and did not exceed 0.25% *v/v*.

*Cell culture studies*. The cell lines T47D, MCF7, BT474, SKBR3, BT549, HS578T, OVCAR8, and SKOV3 and the immortalized non-transformed keratinocyte cell line HACAT was kindly provided by Drs. J. Losada and A. Balmain or acquired in ATCC. All lines were grown in DMEM containing 10% fetal bovine serum (FBS) and were supplemented with 100 U/mL penicillin, 5 mM L-glutamine, 100 μg/mL streptomycin at 37 °C and 5% CO_2_ (Sigma-Aldrich, St. Louis, MO, USA). Carboplatin and cisplatin were purchased from Accord Healthcare (the United Kingdom. MA).

*MTT metabolization assays*. For viability assessment, cell proliferation was assayed by MTT (3-(4, 5-dimethylthiazol-2-yl)-2, 5 diphenyltetrazolium bromide) (Sigma Aldrich). Cell lines were plated at 10,000 cells per well in 48-multiwell plates. 24 h later, the cells were treated at correspondent doses of the drugs for 72 h. After that, the medium was aspirated and phenol red-free DMEM with MTT 0.5 mg/mL was added for 60 min in growth conditions. The medium was removed and MTT crystals were solubilized with 0.5 mL of dimethylsulfoxide (DMSO) (Sigma-Aldrich) and evaluated at an absorbance of 555 nm in a multiwell plate reader.

*Cell cycle and apoptosis assay.* For cell cycle analyses, cell lines were plated at 100,000 cells per well in 6-multiwell plates. 24 h later, wells were treated with an IC_50_ dose of *Ru3* for 24 h. In order to fix the cells, the cells were incubated for 15 min in 70% ethanol. After that, pellets were washed in 2% BSA in PBS and were stained with Propidium iodide/RNAse staining solution for 1 h at 4 °C in dark conditions (Immunostep S.L., Salamanca, Spain).

For apoptosis analysis, the same number of cells were treated for 48 h. Then, adherent and floating cells were washed with PBS and incubated in Annexin V binding buffer (Immunostep S.L.,) for 1 h in the dark with Annexin V and PI staining solution (Immunostep S.L). Percentage of dead cells was determined considering early apoptotic (Annexin V-positive, PI-negative), late apoptotic (Annexin V-positive and PI-positive), and residual necrotic (Annexin V-negative, PI-positive) cells which were included as dead cells in the analysis.

Flow cytometry assays were evaluated in a FACSCanto II flow cytometer (BD Biosciences).

*In vivo studies.* For in vivo toxicity analysis, 12 female BALB/c nu/nu mice of 5 weeks old were treated and weighted twice a week intraperitoneally with *Ru3* or cDDP at a fixed dose of 5mg/kg for up to four treatments. Mice were humanely sacrificed by CO_2_ inhalation in an appropriated chamber. This study was performed under the supervision of the Ethics Committee and veterinary staff of Complejo Hospitalario Universitario de Albacete. Mice were monitored and managed at the animal facility following local legal guidelines.

### 2.7. Statistical Analysis

The in vitro experiments data are the average of three independent experiments performed in triplicate, with error bars showing the standard deviation of the triplicates. To determine significant statistical differences, equal variances Tukey’s test was used with ANOVA, respectively, and to compare the media of experimental groups, Student’s *t*-test was used. The values for the statistical analyses are: *, *p* ≤ 0.05; **, *p* ≤ 0.01; ***, *p* ≤ 0.001.

## 3. Results

### 3.1. Synthesis and Characterization of Precursor Ligands and Ruthenium Compounds

At the outset, a new family of bis-phoshino-amines (PNP) ligands was synthetized. The new PNP ligands were obtained to expand the structure space previously covered and with the aim to generate new ruthenium therapeutic agents [36]. The PNP ligands (*L1*–*L10*) were directly obtained by condensing chlorophosphine with the corresponding amine in the presence of Et_3_N (Figure 2). Due to the abundance of commercial amines and phosphines, this methodology is very versatile to expand ligand sets for different purposes. To optimize the synthetic protocol, the reaction was monitored by ^31^P-NMR. The ligands *L1–L10* were obtained in very good yields upon extraction with toluene. Even though PNP ligands are wide-angle bidentade ligands, mononuclear ruthenium compounds *Ru1–Ru10* were obtained after exposing the corresponding ligand to [RuCl_2_(p-cymene)]_2_ in MeOH (Figure 2). Within a 30 min reaction time, an orange solution was observed and found to contain the mononuclear ruthenium derivative at a very good yield. *Ru0* was obtained following a procedure previously reported [36].

The characterization of *L1*–*L10* and *Ru1*–*Ru10* were carried out by analytical methods, infrared (IR), ultraviolet-visible (UV-Vis), and nuclear magnetic resonance (NMR) spectroscopy. Structural elucidation is depicted in the experimental section and illustrated as representative examples of ligands and compounds in Appendix A. In the IR spectrum of *L1–L10* (solid state), the amine group manifested itself with a broad band around 1020–1250 cm^−1^, while a strong absorption around 1090 cm^−1^ was attributed to the stretching vibration of the P–C moiety. The presence of two signals at 5.7–7.1 ppm and at 90–120 ppm in the ^1^H and ^13^C{^1^H} NMR spectra of the ruthenium complexes, respectively, confirmed the presence of the *p-*cymene ligand in the coordination sphere. As expected, upon coordination to ruthenium, the peaks corresponding to the phenyl rings of the PPh_2_ moiety shifted downfield, as noted in the ^1^H NMR spectrum of complexes *Ru1–Ru10.* Explicitly, the characteristic signals for *L1–L10* in the complexes were downfield-shifted in all cases with regard to those of the free ligands. Moreover, the NMR patterns of the arene resonances supported the asymmetric nature of the resulting complexes. Indeed, the expected ABCD spin system for the *p*-cymene resonances was observed. The IR spectra of *Ru1–Ru10* exhibited a broad band around 300 cm^−1^ for the Ru–Cl [46,47], while a strong absorption in the range 1030–1050 cm^−1^ was attributed to the stretching vibration of the P–C moiety. The UV-Vis absorption spectra of *L1–L10* and *Ru1–Ru10* were recorded in DMSO solutions (10^−5^ M) at 25 °C. *L1–L9* presented broad absorption bands centered at 278–290 nm, while *L10* presented intense and well-defined characteristic bands of quinoline moieties at 358 nm. The absorption spectra of the *Ru1–Ru10* compounds featured a very intense band attributed to π−π* electronic transitions corresponding to the Ru-(*p*-cymene) moiety (λ = 250–304 nm). A weak band is attributable to metal to ligand charge transfer transitions (MLCT) from Ru d orbitals to the π* orbitals of the ligands (304–364 nm), and a shoulder assigned to d−d transitions (364–574 nm) [46].

Single crystals suitable for X-ray diffraction were obtained by slow evaporation of solutions of *Ru3* in CH_2_Cl_2_. Selected bond lengths and angles with estimated standard deviations are compiled in Appendix A and crystallographic refinement parameters are given in Appendix A. Figure 3 shows the ORTEP view of *Ru3* as a representative complex of the series. The structure shows the characteristic “three-legged piano-stool” conformation around the Ru center with a *η*^6^-coodinated *p*-cymene ring [48,49]. Its solid-state structure revealed a distorted tetrahedral geometry with a P-Ru-P chelate angle of 68.42(13)°. The Ru−centroid distance for the *Ru3* complex has a value of 1.770 Å (Appendix A) and is standard compared to similar complexes [36]. The bond distance and angle values are comparable to those reported in the literature for ruthenium analogs [50].

### 3.2. Stability of the Ruthenium Complexes in Solution

The ligands *L1–L10* were not soluble in water. *Ru1*–*Ru10* were air stable and soluble in chlorinated solvents, but not indefinitely stable upon air contact. According to NMR experiments, and as a representative example, *Ru3* completely degraded in chloroform after two weeks, with the orange/yellow solutions progressively turning to green and affording a complicated mixture of species. Qualitative tests determined that *Ru4–Ru8* are slightly soluble in water and very soluble in other polar solvents such as methanol and ethanol. *Ru9–Ru10* were sparingly soluble in water, but very soluble in some organic solvents such as dichloromethane and dimethyl sulfoxide. The enhanced solubility of ruthenium compounds relative to the free ligands could be attributable to the blocking of the hydrogen-bonding acceptor atoms of the ligands after coordination. Regarding the PNP ligand influence, the non-aromatic derivatives gave better solubilities than the aromatic analogs because of the lower hydrophobicity. In any event, *Ru1–Ru10* were soluble enough to enable biological studies in aqueous media and circumvent hypothetical administration problems.

As it happens to many metallodrugs, the ruthenium compounds needed to be dissolved in a mixture of H_2_O:DMSO to perform biological assays which in any case did not exceed 0.25% *v/v* of DMSO. Therefore, the stability of the ruthenium compounds in DMSO-d^6^ and DMSO-d^6^:D_2_O was carried out by NMR monitoring. The set of signals belonging to the starting ruthenium complex did not persist throughout the stability experiments carried out in DMSO-d^6^ (25 °C). Partial release in different extents of the *p*-cymene ligand from *Ru1*–*Ru10* was observed due to the disruptive effect of DMSO (see Appendix A as a representative example in the Appendix A). However, this progressive dissociation is dismissed in the presence of water (Appendix A). The molar ratio between them remained constant in spectra recorded after one week in all the samples. In addition, the respective single set of resonances for the *p*-cymene ligand in the ruthenium compounds were not modified upon the addition of 0.15 mM NaCl. The existence of only one pattern for the ruthenium derivatives suggested ruling out fast chloride dissociation from *Ru1–Ru10*, analogously to that found for RAPTA-C [51]. The mechanism of action of RAPTA-C and its derivatives remains to be fully understood [52]. Although it is assumed that DNA is the target for metallodrugs, evidence suggests that, for certain compounds, proteins are likely involved in the therapeutic effect. It is well known that noncationic complexes underwent aquation to a notable extent, depending on the counterion and the chemical structure of the auxiliary ligand [53]. It is expected for *Ru0*–*Ru10* the formation of the more reactive aqua derivatives from their respective chloride precursors as the activation step for subsequent interaction with the biological targets.

### 3.3. Antiproliferative Efficacy of New Synthetized Ruthenium Compounds Compared to Ru0 and Platins

In a previous study, we reported *Ru0* as the lead PNP-Ru compound by preliminary biochemical and biological studies in different breast cancer subtypes [37]. The results showed that complex *Ru0* is much more effective in promoting in vitro cytotoxic effect on HER2+ and RH+/HER2− breast cancer cell lines than the reference metallodrugs cisplatin, carboplatin, or RAPTA–C [35]. Therefore, a pharmacologic screening using *Ru1–Ru10* were performed in a panel of representative tumor cell lines of each subtype: luminal (RH+/HER2−:T47D and MCF7) and HER2+ (BT474 and SKBR3) (Figure 4A). As can be seen in Figure 4B,C, most ruthenium compounds showed better antiproliferative effects than platinum agents used as controls in all the cancer cell lines evaluated. *Ru2* and *Ru3* showed the highest cytotoxic activity, being *Ru3* the most active compound of all analyzed (see IC_50_ values for T47D, MCF7, BT474, and SKBR3 cells of the most active ruthenium compounds *Ru0*, *Ru2*, and *Ru3* in Table 1). Ligands *L1–L10* did not show significant cytotoxicity with IC_50_ > 1000 (see Appendix A). Importantly, *Ru3* displayed high cytotoxicity with IC_50_ values lower than 1 µM, and was significantly more cytotoxic than cisplatin and carboplatin in the cancer cell lines studied. Notably, *Ru3* was found to be highly cytotoxic in both HER2+ and luminal breast cancer cell lines, suggesting a mode of action independent of the receptor status of the cells. The IC_50_ values observed for *Ru3* were in the range for the most cytotoxic ruthenium complexes reported to date [35,54]. *Ru3* reports very low IC_50_ values, confirming its high potential for breast cancer treatment (see IC_50_ values of *Ru3* for all cell lines in Table 1). Otherwise**,** those ruthenium compounds with aromatic rings in the PNP moiety (*Ru4–Ru10*) did not reduce the cell viability by more than 50% at the concentration tested (Figure 4 and data not shown). These results clearly indicate that the substitution in the PNP moiety can lead to a significant increase in the cytotoxicity of the resulting complexes.

Many factors could be involved in the cytotoxicity of the new ruthenium derivatives such as physical and chemical properties, target affinity, or even plausible interaction with other biomacromolecules. The relatively high activity exhibited by *Ru3* could be somehow related to the stability of the ruthenium complexes in biological media. Appendix A showed the high stability of these compounds in water which rule out any potential relationship between stability and cytotoxicity. On the other hand, the variability in the pharmacological profiles observed might depend on the ability of these compounds to penetrate biological membranes, which in turn is mainly dependent on lipophilicity factors [53]. In this context, calculated logarithmic octanol/water partition coefficients (clogP) for *Ru1–Ru10* were obtained using the software Molinsipiration. The variability of the substitution in the PNP moiety may explain the observed lipophilicity pattern. Appendix A depicts the logP versus %MTTs transformed. A general trend in which the cytotoxicity could correlate with lipophilicity was not observed for these derivatives. In an effort to find a correlation between activity and affinity to proteins, the affinity to HSA of the most active compound of the series, *Ru3*, one of the less active, *Ru8*, and *Ru0* for comparison were obtained by UV-Vis absorption spectroscopy studies (see Appendix A). Unfortunately, a significant difference in the *K*_a_ values was not observed between species which did not facilitate the discovery of any structure–activity relationship.

### 3.4. Ru3 Exerts a G2/M Arrest in Breast Cancer Cell Lines

To further explore the effect of *Ru3* on cell viability, cell cycle analysis after a 24 h incubation period with *Ru3*-IC_50_ was performed (Figure 5) (see representative FACS plots for cell cycle and apoptosis in Appendix A). Due to the promising cytotoxicity results in luminal and HER2+ cell lines, we expanded our research to the three breast cancer subgroups, including the basal-like subtype. As a result, six cell lines (BT474 and SKBR3 as HER2+, T47D and MCF7 as luminal and HS578T, and BT549 as basal-like) were treated with either the vehicle or *Ru3* for 24 h and stained with propidium iodide/RNase solution. As shown in Figure 5, *Ru3* blocked cell cycle progression at the G0/G1 phases in all of the cell lines evaluated except for the HER2-positive subgroup (SKBR3 and BT474).

### 3.5. Ru3 Exerts Apoptotic-Based Cell Death in TNBC

On the basis of the cytotoxic results of *Ru3* in breast cancer cell lines, the mechanism of cell death induction was evaluated. We analyzed *Ru3* ability to induce apoptosis by flow cytometry as previously described [37,53,55,56,57]. The experiments were performed using an incubation time of 48 h, and *Ru3* concentrations corresponding to their respective IC_50_ value. Treatment with *Ru3* was able to significantly increase apoptosis in all the cell lines analyzed, (Figure 6), suggesting that the lead compound clearly induces cell death by apoptosis in a wide range of breast cancer cell lines.

### 3.6. Ru3 Is Less Toxic Than Cisplatin Using In Vitro and in In Vivo Studies

We next evaluated the in vitro effect of *Ru3* on the viability of non-tumoral cells. To this end, we used the non-transformed but immortalized keratinocytes cell line HACAT. Interestingly, *Ru3*, the most potent compound of the series, did not display significant toxicity in non-tumoral cells, showing IC_50_ values which doubled one of the transformed cell lines (Figure 7A and Table 1). The in vivo toxicity of *Ru3* and cisplatin was evaluated by measuring the animal body weight and their behavior. Treatment with *Ru3* and cisplatin was performed using 5 mg/kg intraperitoneal (IP) doses for seven consecutive days. As displayed in Figure 7B, *Ru3* did not induce significant weight loss compared with cisplatin where a 30% weight reduction was observed. No lethality was reported after seven days of treatment with both compounds. This data suggests that a repeated dose of *Ru3* is much better tolerated than cisplatin. On the other hand, in the HACAT cell line, the effects over cell cycle distribution were almost undetectable after 24 h at a 200 nM (Figure 7C), a treatment dose that would cause high toxicity on breast cancer cell lines. Moreover, the effect of that dose over cell death was weaker than the one caused to breast cancer cell lines, despite being statistically significant (Figure 7D).

### 3.7. Ru3 Is Also a Therapeutic Alternative in Ovarian Cancer

TNBC share many biological similarities with ovarian cancer including that both tumors display a high grade of antitumor sensitivity to platinum agents. In this context, we aimed to evaluate the activity of this agent in two well-characterized ovarian cancer cell lines, SKOV3 and OVCAR8, in comparison to cisplatin (cDDP) (Figure 8A,B). To detect differences, we used doses of cisplatin below their IC_50_ (Figure 8C). As can be seen in Figure 8, cisplatin showed modest activity and the action of *Ru3* was much potent at the low doses used.

## 4. Discussion

Breast cancer is a heterogeneous disease where some genomic and clinical characteristics have classified this cancer into several subgroups. Among them, the triple-negative subtype harbors more aggressive behavior, and more therapeutic options are needed due to poor clinical outcomes. In addition, this tumor shares many similarities with ovarian cancer, such as the high grade of genomic instability or the special sensibility to DNA damaging agents including platinum compounds [58]. However, in both cases, after a specific period of time receiving treatment with platinum agents, progression is observed, and additional therapeutic options are needed.

In this context, the rapid development of drug-based metals other than platinum has placed organometallic ruthenium (II) complexes in the spotlight of interest for many drug designers over the last decade. The development of novel compounds in this scenario should fulfill two criteria, first, they must have an optimized toxicity profile, and secondly, their antitumoral activity should remain similar or improved compared with classical platinum agents such as

Cisplatin. In our case, both characteristics are included in the family of new agents described in this report.

Our previous studies allow us to validate PNP-Ru compounds as anticancer drugs for breast cancer therapy establishing a proof of concept for further optimization and characterization of novel agents. In this case, further rational modifications of the molecular structure were required to improve the pharmacological profile of these derivatives. A set of novel PNP ligands was designed to prepare an ample series of ruthenium(II) arene complexes, with the aim of fine-tuning the cytotoxic properties of these types of promising anticancer drugs. Our results suggest that slight modifications in the molecular structure of the PNP-Ru compounds is decisive for tuning its antitumoral activity.

In our work, we position *Ru3* as the lead compound of the series. Three reasons justify this approach: (1) *Ru3* is the most potent antitumoral compound of the series, able to induce cell death without affecting non-transformed cells; (2) its antitumoral action is related to the activation of the intrinsic apoptotic pathway and displays a pharmacological profile that is maintained in cisplatin-resistant cells; (3) it shows less acute toxicity in vitro and in vivo than the reference metallodrug cisplatin and supporting its evaluation in different solid tumors where DNA-damaging agents have a role.

An interesting finding was the fact that *Ru3* induced a profound anti-tumoral effect in TNBC and ovarian cancer cell lines, and that the mechanism of action was mainly mediated by apoptotic cell death. Of note, only a slight increase in G0/G1 was detected, suggesting that the compound is cytotoxic rather than cytostatic. The relevant induction of apoptosis is shared by other metallodrugs such as platinum agents although our drug displayed a safer toxicity profile with a more antitumoral activity [9]. On the other hand, when compared with cisplatin, *Ru3* was less toxic in non-transformed cells and did not induce weight loss in animals after a maintainance dose administration. These findings, in addition to the clear antitumoral activity in vitro of *Ru3* compared with cisplatin or carboplatin, position this compound as a future alternative for clinical development. In addition, *Ru3* displayed a significant antitumoral activity in ovarian cancer, which suggests that these agents can cover a wide range of clinical necessities.

On the other hand, we acknowledge that the full mechanism of action of the compound has not been evaluated in this first study. We could speculate that induction of DNA could be one of those mechanisms, but not the principal, considering the significant induction of cell death observed after its administration. In this context, as an apoptotic inducer agent, combination with other agents acting on complementary molecular vulnerabilities could be exploited, such as the combination with anti-HER2 therapies. Our previous experience reinforces this approach.

## 5. Conclusions

In conclusion, the structure of Ru3 might serve as the basis for the design of more active and less toxic antitumoral compounds and pave the way for the clinical development of these compounds as alternatives for the treatment of tumors. The simple accessible synthesis of this compound and its biological activity makes *Ru3* a good candidate for further evaluation in vivo as a potential chemotherapeutic agent against breast cancer.

## Figures and Tables

**Figure 1 pharmaceutics-13-01559-f001:**
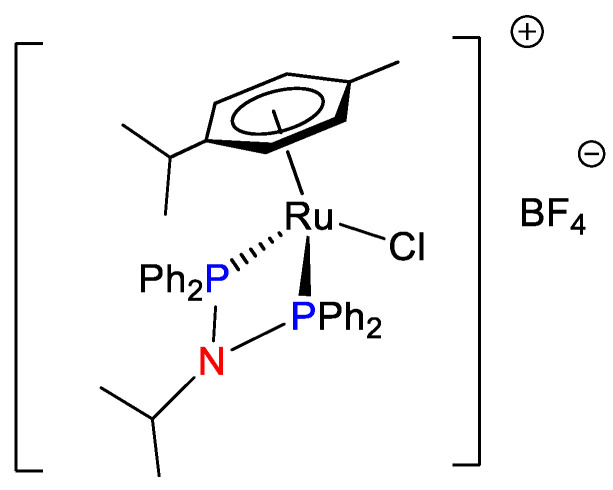
Chemical structure of the lead bis-phosphino-amine compound for breast cancer therapy.

**Figure 2 pharmaceutics-13-01559-f002:**
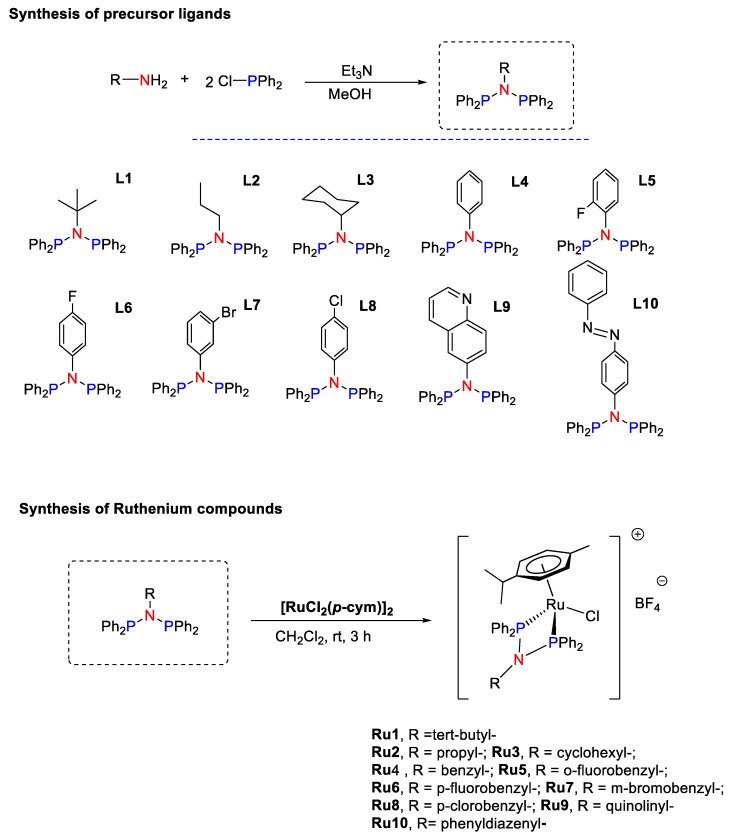
Synthesis of precursor ligands *L1–L10* and ruthenium compounds *Ru1–Ru10*.

**Figure 3 pharmaceutics-13-01559-f003:**
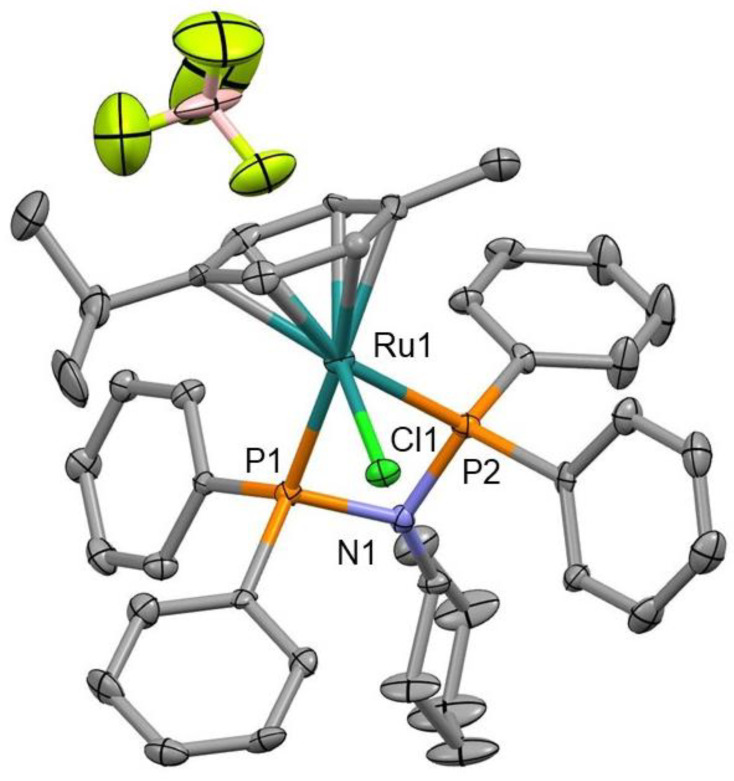
MERCURY plot for the structure of complex *Ru3.* Thermal ellipsoids are given at the 30% probability level. Hydrogen atoms have been omitted for clarity. Selected bond distances (Å), Ru1–Cl1: 2.383(3) Å, Ru1–P1: 2.343(3) Å, Ru1–P2: 2.296(4) Å, N1–P1: 1.717(11) Å, N1–P2: 1.703(11) Å.

**Figure 4 pharmaceutics-13-01559-f004:**
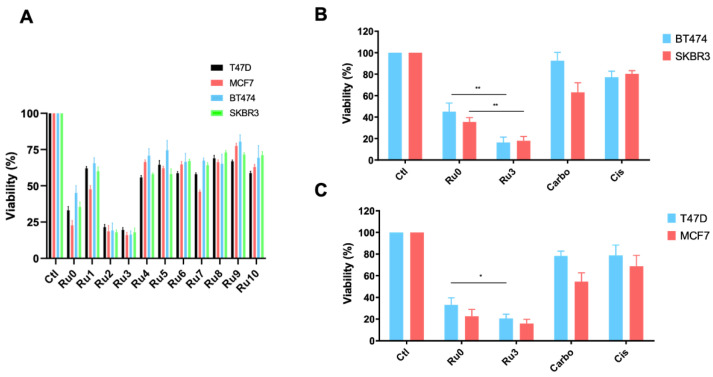
Impact in cell viability of ruthenium compounds at 1 µM. (**A**) Screening of several ruthenium compounds by exploring cell viability in BT474, SKBR3, T47D, and MCF7 breast cancer cell lines for 72 h evaluated using MTTs. *Ru3* displayed the most anti-proliferative activity. The cytotoxic activity was evaluated at 1 µM for 72 h by MTT metabolization between Ru0, carboplatin, and cisplatin in HER2+ (**B**) and luminal (**C**) cell lines. Data are the average +/− standard deviation (SD) of three independent experiments performed in triplicate. To determine significant statistical differences, Student’s *t*-test was used. The values for the statistical analyses are: * *p* ≤ 0.05; ** *p* ≤ 0.01.

**Figure 5 pharmaceutics-13-01559-f005:**
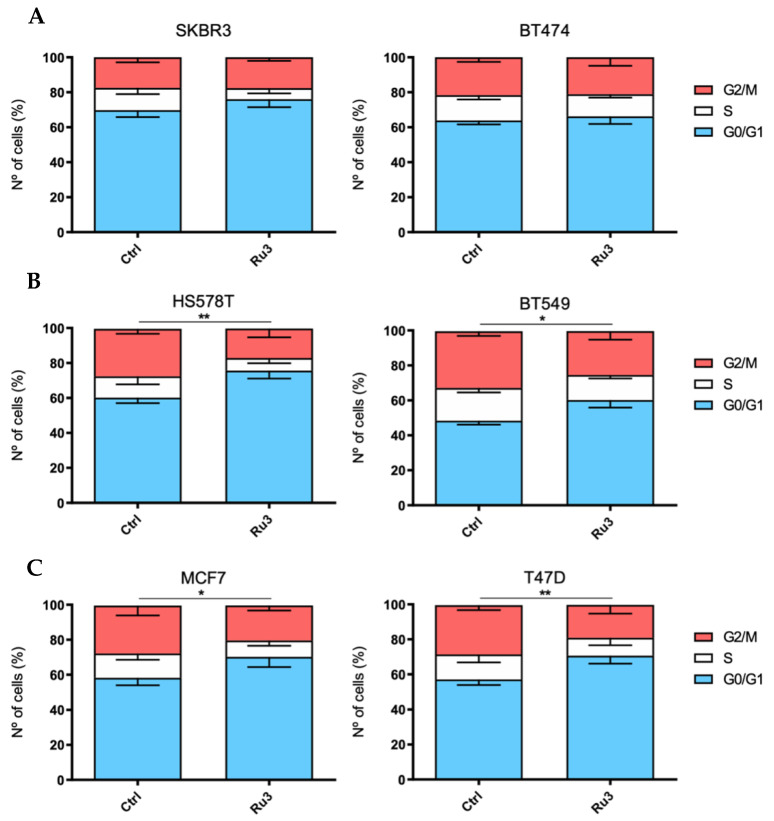
Evaluation of cell cycle impact exerted by *Ru3* in several breast cancer cell line models. Cell cycle progression was evaluated in HER2+ (**A**), basal (**B**), and luminal (**C**) cell lines by propidium iodide staining and cytometer evaluation after 24 h of IC_50_ *Ru3* dose treatment for each cell line. Data are the average +/− standard deviation (SD) of three independent experiments performed in triplicate. To determine significant statistical differences between G0/G1 phase averages, Student’s *t*-test was used. The values for the statistical analyses are: *, *p* ≤ 0.05; **, *p* ≤ 0.01.

**Figure 6 pharmaceutics-13-01559-f006:**
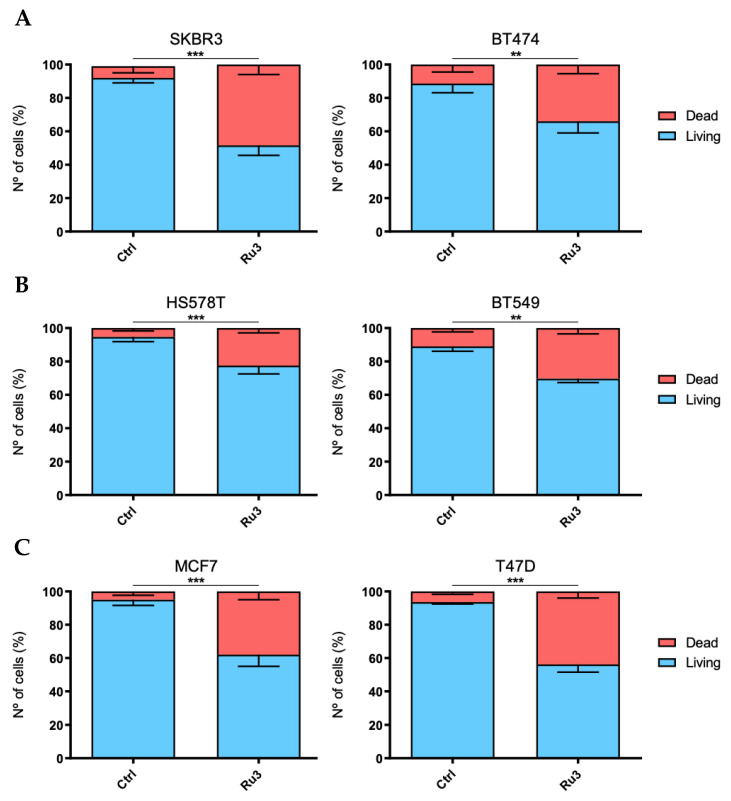
Evaluation of cell death mechanism exerted by *Ru3* in several breast cancer cell line models. Cell death was evaluated in HER2+ (**A**), basal (**B**), and luminal (**C**) cell lines by Annexin V/propidium iodide staining and cytometer evaluation after 48 h using the IC_50_ *Ru3* dose treatment specific each cell line. Data are the average +/− standard deviation (SD) of three independent experiments performed in triplicate. To determine significant statistical differences between living cells groups, Student’s *t*-test was used. The values for the statistical analyses are: *, *p* ≤ 0.05; **, *p* ≤ 0.01; ***, *p* ≤ 0.001.

**Figure 7 pharmaceutics-13-01559-f007:**
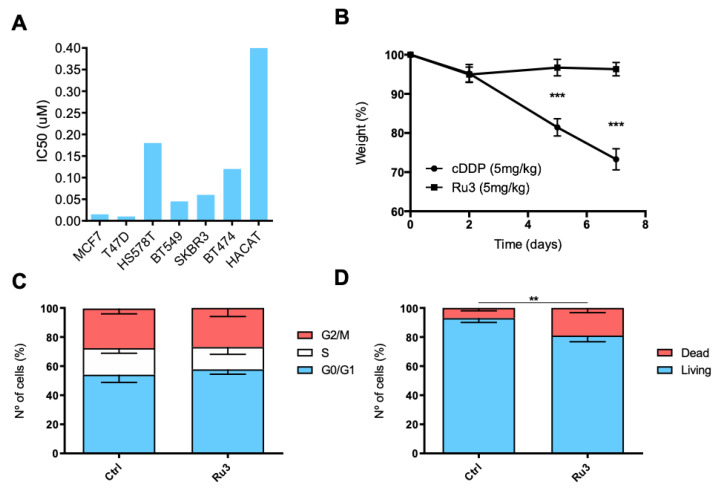
*Ru3* displays less toxicity in non-transformed cells. (**A**) IC_50_ comparison, measured by MTT metabolization, between breast cancer cell lines and HACAT, showing that the IC_50_ of the non-transformed cell line at least doubles the value of the transformed cell lines. (**B**) *Ru3* shows lower toxicity in vivo than platin compounds. Toxicity was evaluated after 7 days of IP treatment with 5 mg kg^−1^ twice a week. *Ru3* (N = 6) exerted significantly less toxicity than cDDP (N = 6) at the 5 mg kg^−1^ with a treatment schedule of two administrations a week. (**C**) The effect of *Ru3* in HACAT cell lines in terms of cell cycle arrest, demonstrating no modifications in any cell cycle phase. (**D**) Cell death was evaluated, and weaker effects were observed when compared to breast cancer cell lines at the dose of 200 nM. Data are the average +/− standard deviation (SD) of three independent experiments performed in triplicate. To determine significant statistical differences between the averages of treatment groups, G0/G1 or living cells groups respectively, Student’s *t*-test was used. The values for the statistical analyses are: *, *p* ≤ 0.05; **, *p* ≤ 0.01; ***, *p* ≤ 0.001.

**Figure 8 pharmaceutics-13-01559-f008:**
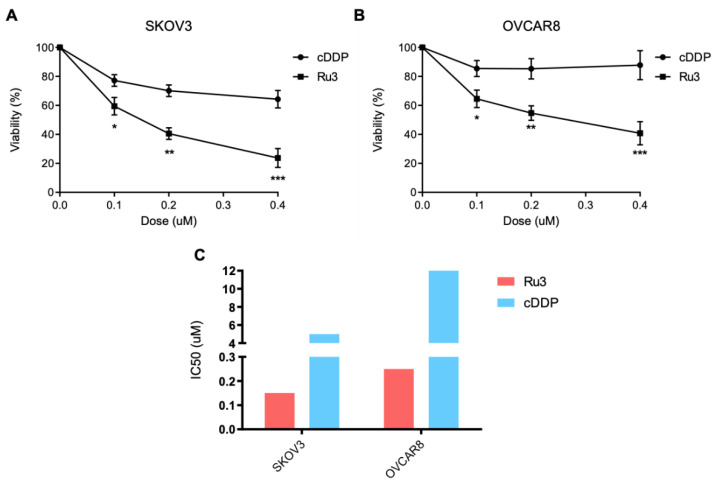
*Ru3* is a putative therapeutic candidate for ovarian cancer treatment. Viability of two ovarian cancer cell lines, OVCAR8 and SKOV3, was assayed by MTT metabolization after 72 h treatment with *Ru3* and cDDP, as a control, showing that *Ru3* developed more toxicity in both cell line models (**A**) and (**B**), as IC_50_ values demonstrate (**C**). Data are the average +/− standard deviation (SD) of three independent experiments performed in triplicate. To determine significant statistical differences, Student’s *t*-test was used. The values for the statistical analyses are: *, *p* ≤ 0.05; **, *p* ≤ 0.01; ***, *p* ≤ 0.001.

**Table 1 pharmaceutics-13-01559-t001:** IC_50_ values of the most active compounds in the different cell lines.

Cell Line	IC_50_ (nM)		
	*Ru0*	*Ru2*	*Ru3*
HACAT	-	-	350
MCF7	300	45	15
T47D	450	60	10
HS578T	-	-	250
BT549	-	-	45
SKBR3	700	80	60
BT474	800	180	120
OVCAR8	-	-	300
SKOV3	-	-	170

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
