# Peer review of "Tuning the Cytotoxicity of Bis-Phosphino-Amines Ruthenium(II) Para-Cymene Complexes for Clinical Development in Breast Cancer"

_pharmaceutics, 2021, doi:10.3390/pharmaceutics13101559_

Round 1

Reviewer 1 Report

The manuscript ”Tuning the Cytotoxicity of bis-phosphino-amines Ruthenium(II) para-cymene complexes for clinical development in Breast Cancer” by Authors: Elena Domínguez-Jurado, Francisco J. Cimas, José Antonio Castro-Osma, Alberto Juan, Agustín Lara-Sánchez, Antonio Rodríguez-Diéguez, Alex Shafir, Alberto Ocaña, carlos Alonso-Moreno, describes synthesis, chemical characterization, water stability and in vitro and in vivo biological evaluation of ten new ligands and corresponding Ru(II) complexes. The manuscript is well written, experiments were adequately performed and all facts are very well presented. The results for chosen Ru3 compound are promising for further biological investigation. Full chemical characterization is correct, but I would only suggest authors, if it is possible, to include mass spectroscopy as well as elemental analysis into chemical characterization part. Also, there were some typo mistakes, Ru3 is not everywhere italic, and check “sp2, sp3” wasn’t always written with superscript.

Taking all into account I recommend this manuscript for the publication in Pharmaceutics after minor revision.

Author Response

We appreciate the Reviewer’s positive view about this study. Following reviewer suggestion, mass spectroscopy and elemental analysis for the ruthenium compounds are included in the revised manuscript. Some representative examples are also included in the Supporting information (Figure S5). We also included equipment and information about the analysis in the materials and methods section. Finally, typo mistakes were corrected. 

Reviewer 2 Report

The manuscript deals with the synthesis and characterization of a family of 10 new diphosphine ligands and their Ru(II) organometallic compounds and their evaluation on different breast cancer cell lines. The mechanism of the lead compound was studied and the toxicity of it was preliminarly evaluated in vivo. The work has almost been properly performed. The topic and the manuscript are of interest in Medicinal Chemistry and are adequate for being published in Pharmaceutics. Nevertheless, the manuscript has to be revised. I suggest major revision.

I have some concerns:

-Elemental analysis are not reported to complete the characterization and identification of the new compounds

-IR bands assignment of those bands put together with the synthesis and characterization under experimental part should indicate type of vibration in each case. In addition, so many figures included for the frequencies of the vibrations values seems not adequate.

-The assignment of 701.269 cm-1 (Ru-Cl) is not clear. Please indicate if it is a stretching. This frequency sems to high for a metal-Cl stretching. Please justify the assignment and if correct add a reference.

-Please put references supporting your assignment of absorption spectra and IR

-About stability in solution of the compounds study it is confusingly reported. NMR show a clear decrease of signals in DMSO which tells about decomposition during the first 24 hours. So the compounds are not very stable. What is the biological relevance of studing stability in DMSO? What do authors hypothesize? What could be the active species?

-Authors wrote: Table S3 in the supporting information showed the high stability of these compounds in water which rule out any potential relationship between stability and cytotoxicity. But Table S3. Shows Stability of Ru1-Ru10 in DMSO-d6 monitoring by 1H-NMR. Authors did not report stability in aqueous solution. Why?

-Why did authors not study the stability in D2O by NMR if they are soluble in water? This stability is the relevant for a prospective drug, and most importantly in biological medium.

-Please correct sentence -Ru3 reports very high IC50 values, confirming its high potential for breast cancer treatment.- IC50 values should be low.

-About activity test Figure 4. Does not show at which dose the compounds are compared

- On Figure 7 C. IC50 comparation, measured by MTT …. The y axis should be IC50 (units) and not dose

-Please put a table with the IC50 values of all the compounds in the different cell lines. These data are lacking and are not clear to obtain from the figures..

-The reported in vivo study is a very preliminar toxicity study. There are alternative studies to test toxicity of compounds not using mice like C. elegans and zebrafish models.

-Under conclusions, the sentence In summary, we describe novel mimetics. is non understandable.

-Why the structure of Ru3 might serve as the basis for the design of more active and less toxic antitumoral agents for the treatment of different solid tumors. Non test for solid tumors has been performed.

Author Response

Comments and Suggestions for Authors

The manuscript deals with the synthesis and characterization of a family of 10 new diphosphine ligands and their Ru(II) organometallic compounds and their evaluation on different breast cancer cell lines. The mechanism of the lead compound was studied and the toxicity of it was preliminarly evaluated in vivo. The work has almost been properly performed. The topic and the manuscript are of interest in Medicinal Chemistry and are adequate for being published in Pharmaceutics. Nevertheless, the manuscript has to be revised. I suggest major revision.

Authors response. We appreciate the Reviewer’s positive view about this study. Please, see below the response to each concern.

I have some concerns:

-Elemental analysis are not reported to complete the characterization and identification of the new compounds

Following reviewer suggestion, elemental analysis is included in the revised manuscript. We also included equipment and information about the analysis in the materials and methods section.

-IR bands assignment of those bands put together with the synthesis and characterization under experimental part should indicate type of vibration in each case. In addition, so many figures included for the frequencies of the vibrations values seems not adequate.

-The assignment of 701.269 cm-1 (Ru-Cl) is not clear. Please indicate if it is a stretching. This frequency sems to high for a metal-Cl stretching. Please justify the assignment and if correct add a reference.

-Please put references supporting your assignment of absorption spectra and IR.

IR bands assignment have been revised, indicating type of vibration in each case. We would like to apologize for those mistakes reported for several frequencies in our original manuscript. Finally, some references to support the assignment were also included in the revised manuscript. 

-About stability in solution of the compounds study it is confusingly reported. NMR show a clear decrease of signals in DMSO which tells about decomposition during the first 24 hours. So the compounds are not very stable. What is the biological relevance of studing stability in DMSO? What do authors hypothesize? What could be the active species?

-Authors wrote: Table S3 in the supporting information showed the high stability of these compounds in water which rule out any potential relationship between stability and cytotoxicity. But Table S3. Shows Stability of Ru1-Ru10 in DMSO-d6 monitoring by 1H-NMR. Authors did not report stability in aqueous solution. Why?

-Why did authors not study the stability in D2O by NMR if they are soluble in water? This stability is the relevant for a prospective drug, and most importantly in biological medium.

We really appreciate this observation reported by the reviewer and we have modified the section about stability of the ruthenium compounds to avoid any misunderstanding. As it happens to many metallodrugs, the ruthenium compounds need to be dissolved in a mixture H2O:DMSO to perform biological assays. The maximal concentration used for the biological assays was 50 μM for all the ruthenium compounds, whereas platin derivatives were tested up to 100 μM. The same volume of solvent was added to the control and did not exceed 0.25% v/v. This information is included in the material and methods section. This is the reason why we had to study the stability of compounds in DMSO-d6 and DSMO-d6/D2O, as it is usually carried out for other authors (see DOI: 10.1039/c7dt03385k as a representative example).  The disruptive effect of DMSO is likely to be responsible for the partial release of the p-cymene ligand ((a) J. Organomet. Chem., 2011, 696, 2485–2490; Chem. Sci., 2015, 6, 2449–2456.) Successfully, we observed how these compounds are highly stable as soon as water was added to DMSO. Indeed, arene dissociation from Ru(II) complexes in water is generally not observed (Organometallics, 2012, 31, 7031–7039; J. Organomet. Chem., 2011, 696, 999–1010)

-Please correct sentence -Ru3 reports very high IC50 values, confirming its high potential for breast cancer treatment.- IC50 values should be low.

The sentence has been replaced by -Ru3 reports very low IC50 values, confirming its high potential for breast cancer treatment.-

-About activity test Figure 4. Does not show at which dose the compounds are compared

We agree with the reviewer and apologize for the mistake. We have rewritten Figure 4 legend as -Figure 4. Impact in cell viability of ruthenium compounds at 1 µM.-

- On Figure 7 C. IC50 comparation, measured by MTT …. The y axis should be IC50 (units) and not dose

Please put a table with the IC50 values of all the compounds in the different cell lines. These data are lacking and are not clear to obtain from the figures.

According to Reviewer 2 suggestion, IC50 (µM) replaces Dose (µM) Moreover, Table S5, included in the Supporting Information, shows IC50 values for each cell line.

-The reported in vivo study is a very preliminar toxicity study. There are alternative studies to test toxicity of compounds not using mice like C. elegans and zebrafish models.

Preliminary toxicity studies were carried out before starting in vivo studies. Seeing the toxicity of the lead compound, it is mandatory to plan further in vivo studies. Unfortunately, we do not have facilities to carry out the study on C. Elegants or zebrafish models, which is a very interesting suggestion. Nonetheless, assaying toxicity in a mammal context like the mice model mimics better human physiology, especially in tissues like the mammary gland or ovaries where hormonal regulation and tissue microenvironment are key elements.

-Under conclusions, the sentence In summary, we describe novel mimetics. is non understandable.

-Why the structure of Ru3 might serve as the basis for the design of more active and less toxic antitumoral agents for the treatment of different solid tumors. Non test for solid tumors has been performed.

We agree with his/her comments, and we have rewritten the conclusion section to avoid misunderstandings.

Reviewer 3 Report

This manuscript described Ru3 as a potential drug treatment for breast cancer.

More information is needed to clarify their claims in regards to its biological activity.

  1. In Figures 4-7, on all in vitro experiments, the authors did not clarify if these were repeated. All in vitro experiments should be repeated if not done so, to confirm that their results are valid.
  2. In all figure legends, number of experiments (n=) and error bars (SD or SEM) should be clearly stated. Statistical test used should also be included.
  3. In Figure 4, the dose used for their screening studies was not stated, so unsure if their comparisons are appropriate.
  4. Before Figure 5, it would be important to show how the IC50 values for Ru3 on all the cell lines were generated  (ie viability dose curves).
  5. Representative FACS plots for cell cycle and apoptosis is required to clarify the data generated.
  6. The authors claim that there was cell cycle arrest observed after 24 hours, it would be interesting to see if this is exacerbated at 48 hours. This could then be related to the apoptosis data generated.
  7. For Figures 5-7, it is not clear what statistically significant comparisons were made.
  8. Apoptosis was concluded from Figure 6, however, the authors did not state what 'dead' meant here. Was it all Annexin V+ or PI+? Gating strategy would help. Also assays to determine any increase in Caspase 3/7 activation would be desirable.
  9. Other than weightloss, were there other measurements made to ensure that there were no toxic side effects observed? Most animal welfare sheets would also include ruffled coat and reluctance to move, etc... The authors can potentially make a comment on this.

Author Response

Comments and Suggestions for Authors

This manuscript described Ru3 as a potential drug treatment for breast cancer.

More information is needed to clarify their claims in regards to its biological activity.

Author response. Please, see below the response to each concern.

  1. In Figures 4-7, on all in vitro experiments, the authors did not clarify if these were repeated. All in vitro experiments should be repeated if not done so, to confirm that their results are valid.

We agree with the reviewer and apologize for the omission. Results were obtained as the average of three independent experiments performed in triplicate. A paragraph to clarify this fact was added into the statistical analysis section of Materials and Methods.

  1. In all figure legends, number of experiments (n=) and error bars (SD or SEM) should be clearly stated. Statistical test used should also be included.

We agree with the reviewer again. We have reformulated the Statistical Analysis section as follows: - In vitro experiments data are the average of three independent experiments performed in triplicate, with error bars showing standard deviation of the triplicates. To determine significant statistical differences, to equal variances Tukey's test was used with ANOVA respectively, and to compare the media of experimental groups, Student t test was used. The values for the statistical analyses are: *, p ≤ 0.05; **, p ≤ 0.01; ***, p ≤ 0.001.-

In addition, we have added a sentence at the end of each figure to clarify the statistical analysis of the data.

  1. In Figure 4, the dose used for their screening studies was not stated, so unsure if their comparisons are appropriate.

We apologize for this omission also noticed by another reviewer. We have rewritten Figure 4 legend as: - Figure 4. Impact in cell viability of ruthenium compounds at 1 µM. A. Screening of several ruthenium compounds by exploring cell viability in BT474, SKBR3, T47D and MCF7 breast cancer cell lines for 72h evaluated using MTTs. Ru3 displayed the most anti-proliferative activity. The cytotoxic activity was evaluated at 1 µM for 72h by MTT metabolization between Ru0, carboplatin and cisplatin in HER2+ (Fig. 4B) and luminal (Fig. 4C) cell lines. Data are the average +/- standard deviation (SD) of three independent experiments performed in triplicate. To determine significant statistical differences, Student t test was used. The values for the statistical analyses are: *, p ≤ 0.05; **, p ≤ 0.01; ***, p ≤ 0.001.

4.Before Figure 5, it would be important to show how the IC50 values for Ru3 on all the cell lines were generated  (ie viability dose curves).

We agree with the reviewer. According to this suggestion, table S5 showing IC50 values for each cell line is included in the Supporting Information.

  1. Representative FACS plots for cell cycle and apoptosis is required to clarify the data generated.

We agree with the referee that clarifying flow cytometry data can improve manuscript quality. Representative FACS plots for cell cycle and apoptosis are depicted in Figure S9 in the Supporting Information. Furthermore, following his/her suggestions, we have rewritten the Materials and Methods section regarding cell cycle and apoptosis as follows:

Cell cycle and apoptosis assay. For cell cycle analyses, cell lines were plated at 100.000 cells per well in 6-multiwell plates. 24 hours later, wells were treated with IC50 dose of Ru3 for 24h. In order to fix the cells, those were incubated for 15 minutes in 70% ethanol. After that, pellets were washed in 2% BSA in PBS and were stained with Propidium iodide/RNAse staining solution for 1 h at 4 °C in dark conditions (Immunostep S.L., Salamanca, Spain).

For apoptosis analyses, the same number of cells were treated for 48 h. Then, adherent and floating cells were washed with PBS and incubated in Annexin V binding buffer (Immunostep S.L.,) for 1 hour in the dark with Annexin V and PI staining solution (Immunostep S.L). Percentage of dead cells was determined considering early apoptotic (Annexin V-positive, PI-negative), late apoptotic (Annexin V-positive and PI-positive) and residual necrotic (Annexin V-negative, PI-positive) cells which were included as dead cells in the analysis.

Flow cytometry assays were evaluated in a FACSCanto II flow cytometer (BD Biosciences).

6.The authors claim that there was cell cycle arrest observed after 24 hours, it would be interesting to see if this is exacerbated at 48 hours. This could then be related to the apoptosis data generated.

We ruled out performing apoptosis cytometry evaluation at 48h due to the high toxicity exerted by the compound at 24 hours. In our expertise, we would rather observe the differences in cell cycle before causing damage to the cells. This is an interesting idea suggested by the reviewer for future studies to understand if there is a relationship between cell cycle arrest and apoptosis.

  1. For Figures 5-7, it is not clear what statistically significant comparisons were made.

Thank you for noticing this. We have made it clear by adding the following sentences for each figure legend:

Fig. 5: -To determine significant statistical differences between G0/G1 phase averages, Student t test was used.-

Fig.6: -To determine significant statistical differences between living cells groups, Student t test was used.-

Fig. 7. -To determine significant statistical differences between the averages of treatment groups, G0/G1 or living cells groups respectively, Student t test was used.-

8.Apoptosis was concluded from Figure 6, however, the authors did not state what 'dead' meant here. Was it all Annexin V+ or PI+? Gating strategy would help. Also assays to determine any increase in Caspase 3/7 activation would be desirable.

About the gating strategy, the original submission of our manuscript already included an explanation of cell death population gating, that was included in the Material and Methods section. However we have now modified the text in the following manner: -For apoptosis analyses, the same number of cells were treated for 48 h. Then, adherent and floating cells were washed with PBS and incubated in Annexin V binding buffer (Immunostep S.L.,) for 1 hour in the dark with Annexin V and PI staining solution (Immunostep S.L). Percentage of dead cells was determined considering early apoptotic (Annexin V-positive, PI-negative), late apoptotic (Annexin V-positive and PI-positive) and residual necrotic (Annexin V-negative, PI-positive) cells which were included as dead cells in the analysis.-

Finally, we did not include the caspase 3/7 evaluation as modifications in apoptosis were clearly observed with the mentioned experiments.

  1. Other than weightloss, were there other measurements made to ensure that there were no toxic side effects observed? Most animal welfare sheets would also include ruffled coat and reluctance to move, etc... The authors can potentially make a comment on this.

We agree with the reviewer comment. Weight lost is not the only parameter that reflects side effects, indeed other parameters can be evaluated. Unfortunately, we did not quantify these parameters in the in vivo study and, therefore, we are not able to include them in the revised manuscript. Moreover, weight loss is the combination of both behavioural (reducing food amount eat) and physiological changes due to chemotherapy, being a good readout of the impact of the drug in the wellbeing of the animals.

Round 2

Reviewer 2 Report

Authors have addressed some of my concerns. I still have concerns and therefore the manuscript is still not suitable for publication in its current form.

Too many figures are included in the IR frequencies values. For example 2961.201 should be 2961. Please correct all.

In my previous review I asked to please put a table with the IC50 values of all the compounds in the different cell lines. Authors just put the data por Ru3 in Table S5. At least this table but including free ligands should be included in the main text. Data from the figures are not easily accessible to the reader. In addition, a Table has to be included under Supplementary showing IC50 values for all compounds and ligands in all cell lines. The reader has to have available the data in order to compare the different ruthenium compounds and the free ligands.

More comments related to the possible active species are needed.

Author Response

Authors have addressed some of my concerns. I still have concerns and therefore the manuscript is still not suitable for publication in its current form.

Too many figures are included in the IR frequencies values. For example 2961.201 should be 2961. Please correct all.

Author response. Figures regarding to IR frequencies values have been corrected.

In my previous review I asked to please put a table with the IC50 values of all the compounds in the different cell lines. Authors just put the data por Ru3 in Table S5. At least this table but including free ligands should be included in the main text. Data from the figures are not easily accessible to the reader. In addition, a Table has to be included under Supplementary showing IC50 values for all compounds and ligands in all cell lines. The reader has to have available the data in order to compare the different ruthenium compounds and the free ligands.

Author response. Following reviewer suggestion, a new Table with the IC50 values for Ru3, Ru2 and Ru0  is included in the main text  to have available  the data depicted in the figures. Moreover, Table S5 in the Supporting Information collects IC50 values obtained in this study.

More comments related to the possible active species are needed.

Author response. The expected active species for this family of RAPTA derivatives have postulated in the revised manuscript.

Reviewer 3 Report

The authors have addressed my concerns .

Author Response

The authors have addressed my concerns .

Author response. We appreciate the Reviewer’s positive view about this study.

Round 3

Reviewer 2 Report

The authors have now addressed my last concerns. The manuscript is suitable for publication.